# GOCM: Single-Step Graph Outlier Synthesis via Origin Consistency Model

Yifan Li [1]   Zhihui Wang [1]   Changmiao Wang [2]   Guangxiao Ma [1]   Peng Zhang [1]

## Abstract

Supervised Graph Outlier Detection has long been constrained by severe class imbalance, and although recent diffusion-based augmentation methods have improved sample quality, their practical utility is hindered by the high computational costs of multi-step iterative sampling and the stochasticity of the generation process. To overcome these bottlenecks, we propose Graph Outlier Synthesis via Origin Consistency Model (GOCM), a single-step graph outlier synthesis framework based on a consistency model. Theoretically, we pioneer the Origin Consistency (OC) mechanism by employing an "Interval-based Origin Inference" strategy, which mathematically derives a direct mapping from the noise trajectory to the data origin, achieving robust and efficient single-step sample generation. Architecturally, to address the complexity of heterogeneous graphs containing multiple relations, we design the Multi-input Variational Graph Auto-Encoder (MiVGAE), which decouples intricate structures via relation-level message passing and cross-relation fusion, mapping them into a unified latent space, from which GOCM synthesizes high-quality outlier nodes. Extensive experiments on multiple real-world datasets demonstrate that GOCM achieves superior detection performance with significantly improved generation efficiency. The source code is publicly available at: https://github.com/1312267710/gocm.

## 1. Introduction

Graph Outlier Detection (GOD) has become indispensable in high-stakes domains such as fraud detection (Huang et al., 2022; Pan et al., 2025), spam detection (Dou et al., 2020b),

anti-money laundering (Weber et al., 2019; Tang et al., 2023) and abnormal pattern detection on social networks (Zhang et al., 2025). While significant progress has been made using structural anomalies, attribute deviations, and latent geometric properties (Ding et al., 2019; Fan et al., 2020; Grover et al., 2025), the field remains fundamentally challenged by severe class imbalance. In real-world scenarios like financial networks, outliers are extremely scarce (e.g., only 1.3% in DGraph (Huang et al., 2022)), causing loss functions to bias toward the majority class and limiting model generalization (Tang et al., 2022; Liu et al., 2025).

To mitigate this, researchers have employed data-level strategies like under-sampling (Dou et al., 2020a) or feature shuffling (Liu et al., 2022), but these often fail to capture the intrinsic complexity of graph data. Recently, generative approaches, particularly Diffusion Models (DM) (Ho et al., 2020) and Latent Diffusion Models (LDM) (Rombach et al., 2022), have shown promise in synthesizing high-quality samples. Some works leverage Latent Diffusion Models to generate pseudo-anomalous graph (Cai et al., 2025), while others have directly achieved effective synthesis of graph outliers (Liu et al., 2025). For more details about the relevant methods, please refer to Appendix A. However, LDM-based paradigms face two critical bottlenecks: (1) Inefficiency: The multi-step iterative sampling process incurs high computational cost and uncertainty during large-scale data augmentation; (2) Heterogeneity: Existing methods perform poorly in modeling the multiple types of relationships (edges) that exist in complex financial or social networks.

To break through these limitations, we propose Graph Outlier Synthesis via Origin Consistency Model (GOCM), a novel framework for efficient, single-step outlier synthesis on heterogeneous graphs. At its core, GOCM introduces an Origin Consistency (OC) mechanism that replaces the iterative process or Consistency Models (CM) with a direct mapping from noise to the data origin (see Fig. 1). By deriving the intrinsic relationship between trajectory segments and the origin, OC establishes a principled training foundation that bypasses complex discretized curricula (Song et al., 2023; Geng et al., 2025). Furthermore, to handle heterogeneity, we design a Multi-input Variational Graph Auto-Encoder (MiVGAE) that decouples structural encoding from generation, allowing outliers to be synthesized directly in a relation-aware latent space.

[1]College of Computer Science and Engineering, Shandong University of Science and Technology, Qingdao, China [2]Shenzhen Research Institute of Big Data, Shenzhen, China. Correspondence to: Zhihui Wang <zh_wang@sdust.edu.cn>.

*Proceedings of the $43^{rd}$ International Conference on Machine Learning*, Seoul, South Korea. PMLR 306, 2026. Copyright 2026 by the author(s).

In summary, our main contributions are as follows:

- We first introduce consistency model to the task of graph anomaly detection, and propose a consistency model-based graph anomaly data augmentation paradigm.

- We propose the OC Model, which fundamentally improves upon standard consistency models by reducing training difficulty and enabling efficient single-step synthesis for graph outliers.

- We design the MiVGAE framework to effectively handle multi-relational data, enabling it to be applicable to large-scale heterogeneous graph applications.

- Extensive experiments demonstrate that GOCM consistently matches or surpasses state-of-the-art methods on multiple benchmarks while substantially reducing computational costs.

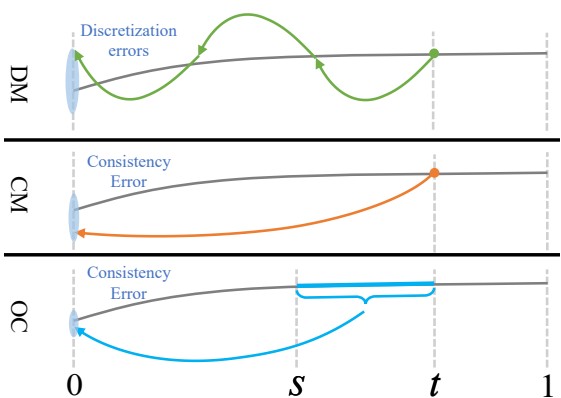

*Figure 1.* **Sampling comparisons among three models.** (Top) Diffusion Models (DMs) utilize discrete time steps for sampling, which incurs discretization errors. (Middle) Standard Consistency Models (CM) focuses on mapping an individual point $\mathbf{x}_t$ along the trajectory to the data origin $\mathbf{x}_0$, which still incurs a consistency error. (Bottom) The proposed OC mechanism utilizes the local trajectory segment (blue line) within the time interval $[s, t]$ to directly infer the origin $\mathbf{x}_0$, thereby reducing the consistency error.

## 2. Preliminaries

### 2.1. Graph Outlier Detection

The Graph Outlier Detection task is designed to train a graph anomaly detector utilizing a labeled training set $D_{\text{train}} = \{(\mathcal{G}, \mathbf{y})\}$ ($\mathcal{G}$ is graph, $\mathbf{y} \in \{0, 1\}$ represents the node label). A critical issue is that, due to the class imbalance problem, the number of anomalous nodes ($y_i = 1$) in $D_{\text{train}}$ is substantially smaller than that of normal nodes ($y_i = 0$).

This work aims to address this by augmenting the training set $D_{\text{train}}$ through the synthesis of high-quality anomalies.

Specifically, our objective is to learn a conditional generative model that produces a set of realistic and diverse synthetic anomalies, denoted $D_{\text{syn}}$. The final augmented training set, $D_{\text{aug}} = D_{\text{train}} \cup D_{\text{syn}}$, is then used to train the downstream anomaly detector.

### 2.2. Consistency Models

Consistency Models are efficient single-step generative models evolved from Denoising Diffusion Probabilistic Models (DDPM) and Probability Flow Ordinary Differential Equations (PF-ODE) (Song et al., 2020). DDPM defines a trajectory $\mathbf{x}(t)$ that maps the data distribution $\mathbf{x}_0 \sim p_{\text{data}}$ to the prior distribution $\epsilon \sim p_{\text{prior}}$. Without loss of generality, we consider the continuous time interval $t \in [0, 1]$.

The central idea of CM is to train a neural network $\mathcal{C}_\theta(\cdot, \cdot)$ to possess the Consistency Property: the outcome of mapping any two arbitrary points, $\mathbf{x}_t$ and $\mathbf{x}_{t'}$, sampled along the same PF-ODE trajectory back to the original data $\mathbf{x}_0$, must be identical. This property is defined as:

$$\mathcal{C}_\theta(\mathbf{x}_t, t) = \mathcal{C}_\theta(\mathbf{x}_{t'}, t') = \mathbf{x}_0, \quad \text{for } \forall\, t, t' \in [0, 1], \quad (1)$$

which is designed to predict the corresponding original data point $\mathbf{x}_0$.

The CM model is trained by minimizing the following Consistency Loss $\mathcal{L}_{\text{CM}}$:

$$\mathcal{L}_{\text{CM}} = \mathbb{E}\left[\rho(t, t') || \mathcal{C}_\theta(\mathbf{x}_{t'}, t') - \mathcal{C}_\theta(\mathbf{x}_t, t) ||^2\right], \quad (2)$$

where $t$ and $t'$ are adjacent time steps sampled along the same trajectory $\mathbf{x}(t)$, and $\rho(t, t')$ is a weighting function.

## 3. Methodology

The proposed GOCM constitutes a novel and efficient single-step generative framework, designed to mitigate the class imbalance issue prevalent in Graph Outlier Detection.

### 3.1. Framework Overview

As illustrated in Fig. 2, GOCM employs a two-stage approach, specifically designed to decouple the representation encoding of the graph structure from the efficient data generation process:

**Latent Space Encoding:** We design and utilize a MiVGAE to map the graph to a low-dimensional, decoupled latent space $\mathbf{z}$, yielding the latent representation $\mathbf{z} \sim q(\mathbf{z}|\mathcal{G})$. This initial step aims to capture the structural and feature priors of the graph.

**OC Model Learning:** Subsequently, we train an OC Model $g_\theta(\mathbf{x}_t, t, s, \mathbf{c})$. Bridging the two stages, we set the data origin $\mathbf{x}_0$ to be the latent representation $\mathbf{z}$. Operating in the

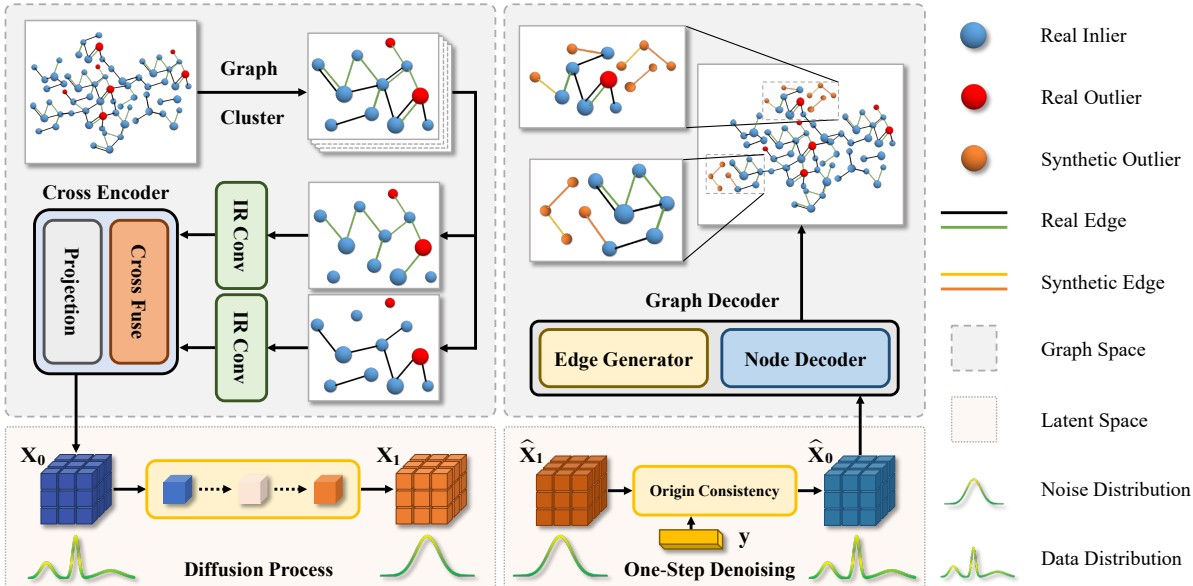

*Figure 2.* **The overall framework of GOCM.** (Left) The MiVGAE encodes the heterogeneous graph structure. It utilizes Intra-Relation Convolution (IR Conv) and Cross-Relation Encoding (Cross Encoder) to fuse multi-relational features into the latent representation $\mathbf{x}_0$. (Right) The Graph Decoder employs the Node Decoder to reconstruct node attributes and the Edge Generator to predict the existence and types of synthetic edges. (Bottom) The OC model executes a single-Step Denoising process that directly synthesizes outlier representations ($\hat{\mathbf{x}}_0$) from noise ($\hat{\mathbf{x}}_1$), conditioned on the label $\mathbf{y}$.

latent space, this model transforms simple Gaussian noise (i.e., $\mathbf{x}_1$, the final sample of the forward process) into the target data distribution $\hat{\mathbf{x}}_0$, thereby simulating the real data distribution $\mathbf{x}_0$. Here, $t$ and $s$ represent different time steps, and $\mathbf{c}$ represents the conditioning information provided by MIVGAE.

**Decoding to Graph Space:** Finally, the generated high-fidelity latent representations of the graph anomalies are decoded back into the original graph space using the decoder. This synthetic data is then used to augment the original graph for training the downstream anomaly detector.

### 3.2. Origin Consistency

The primary optimization objective of standard Consistency Models is Consistency of Network Outputs, which enforces inputs sampled along the same trajectory to map to the identical endpoint. However, as noted by (Geng et al., 2025), this type of consistency is merely an artificial constraint imposed upon the network's behavior, relying on empirical rules including a carefully designed discretization curriculum. This approach not only increases training complexity and makes it prone to error accumulation but also causes the training procedure to be extremely sensitive to hyperparameters.

To optimize the training paradigm of generative models at a more fundamental level, we design the OC function based on the geometry of linear probability flow. Unlike standard CMs, OC aims to embed the temporal interval directly into

the model's underlying theoretical dependencies to learn the trajectory's "origin" $\mathbf{x}_0$. This is a modeling principle we term "Interval-based Origin Inference". We define a conditional function $g_\theta(\mathbf{x}_t, t, s, \mathbf{c})$. For brevity, we denote the conditional function $g_\theta(\mathbf{x}_t, t, s, \mathbf{c})$ as $g_\theta(\mathbf{x}_t, t, s)$ or simply $g_\theta$, omitting the conditioning variable $\mathbf{c}$ and other explicit arguments when the context is clear. This function is designed to directly map an arbitrary point $\mathbf{x}_t$ on the flow trajectory to the unique initial state $\mathbf{x}_0$ of that same trajectory. Specifically, $t$ and $s$ serve as temporal reference points defining the mapping interval $[s, t]$. The core mechanism of this function is to recover the initial state $\mathbf{x}_0$ from any local trajectory segment. In particular, the parameterized function $g_\theta$ encapsulates a direct mapping from an arbitrary noisy state $\mathbf{x}_t$ to its corresponding origin $\mathbf{x}_0$. OC captures the intrinsic information of the trajectory by analyzing the dynamics of $\mathbf{x}_t$ over the time interval $[s, t]$. This is achieved through the integration of all instantaneous velocity vectors along the linear flow path, ensuring that the inferred origin remains consistent with the deterministic probability flow.

To establish a physically consistent generation process, we adopt the linear interpolation modeling of ReFlow (Liu et al., 2023), which defines a linear trajectory from the data $\mathbf{x}_0$ to noise $\mathbf{x}_1 = \epsilon \sim p_{\text{prior}}$ over the time interval $t \in [0, 1]$. This offers distinct advantages in path modeling over SDE paths:

$$\mathbf{x}_t = (1 - t)\mathbf{x}_0 + t\mathbf{x}_1. \tag{3}$$

Here, the instantaneous velocity is expressed as $\mathbf{v}_t =$

$\mathbf{x}_1 - \mathbf{x}_0 = \frac{\mathbf{x}_t - \mathbf{x}_0}{t}$. This linear geometry provides the analytical basis for our subsequent consistency derivation. Leveraging this property, we formulate a function $g(\mathbf{x}_t, t, s)$ by integrating the estimated velocity field. Instead of directly modeling the abstract score function, we explicitly utilize the time-normalized drift term:

$$g(\mathbf{x}_t, t, s) = \mathbf{x}_t + \frac{t}{t - s} \int_t^s \frac{\mathbf{x}_u - \mathbb{E}[\mathbf{x}|\mathbf{x}_u]}{u} du. \quad (4)$$

Here, the term $\frac{\mathbf{x}_u - \mathbb{E}[\mathbf{x}|\mathbf{x}_u]}{u}$ represents the velocity estimate at time $u$. Integrating this velocity yields the robust estimation of the data origin. Although this integral form shares structural similarities with the parameterization used in CTM (Kim et al., 2024), our derivation is grounded in the geometry of linear probability flows rather than standard diffusion dynamics.

However, the complex integral form of $g(\mathbf{x}_t, t, s)$ is not computationally practical, and the term $\mathbb{E}[\mathbf{x} \mid \mathbf{x}_t]$, defined as a theoretical denoiser, is difficult to obtain directly. Instead, we propose to learn a parametric mapping $g_\theta(\mathbf{x}_t, t, s)$ via a neural network that approximates this trajectory integration. To derive $g_\theta(\mathbf{x}_t, t, s)$, we perform a mathematical derivation based on $g(\mathbf{x}_t, t, s)$, and the detailed derivation is provided in Appendix B. This results in a computationally tractable approximation of $\mathbb{E}[\mathbf{x} \mid \mathbf{x}_t]$ known as the parameterized denoiser $f_\theta(\mathbf{x}_t, t, s)$. This function utilizes $g_\theta(\mathbf{x}_t, t, s)$ and its derivatives to account for the local flow dynamics, as shown below:

$$f_\theta(\mathbf{x}_t, t, s) = \mathbf{x}_t + \frac{s}{t}(g_\theta - \mathbf{x}_t) + (t - s) \left( \frac{\partial g_\theta}{\partial t} + \left( \frac{\partial g_\theta}{\partial \mathbf{x}_t} - \mathbf{I} \right)(\mathbf{x}_1 - \mathbf{x}_0) \right). \quad (5)$$

Naturally, we can substitute this denoiser $f_\theta(\mathbf{x}_t, t, s)$ into the consistency loss:

$$\mathcal{L}(\theta) = \mathbb{E}\left[ \|\mathbf{x}_0 - f_\theta(\mathbf{x}_t, t, s)\|_2^2 \right]. \quad (6)$$

After transformation, we obtain a function for training $g_\theta$:

$$\mathcal{L}(\theta) = \mathbb{E}\left[ \left( \frac{s}{t} \right)^2 \left\| g_\theta - \left( \frac{t}{s}\mathbf{x}_0 - \frac{\mathbf{x}_t(t-s)}{s} - \frac{t^2}{s} \left( 1 - \frac{s}{t} \right) \left( \frac{\partial g_\theta}{\partial t} + \left( \frac{\partial g_\theta}{\partial \mathbf{x}_t} - \mathbf{I} \right)(\mathbf{x}_1 - \mathbf{x}_0) \right) \right) \right\|_2^2 \right]. \quad (7)$$

This yields a favorable result with a single and explicit minimization objective. Once the parameters are optimized to minimize the above-mentioned expression, the resulting model will fully satisfy the expected design specifications, namely a generative model theoretically capable of one-step synthesis.

To explicitly learn the interval-based mapping without relying on external pre-trained models, we formulate the training process as a regression problem.

**Model Prediction:** The network output $u$ serves as the estimated origin derived from the current noisy state $\mathbf{x}_t$. It is defined as:

$$u = g_\theta(\mathbf{x}_t, t, s, \mathbf{c}). \quad (8)$$

Our goal is to optimize the parameters $\theta$ such that $g_\theta$ accurately predicts the unique initial state $\mathbf{x}_0$ by encapsulating the trajectory evolution over the interval $[t, s]$.

**Regression Target:** We construct a self-consistent regression target $u_{\text{tgt}}$ derived directly from the governing dynamics of the probability flow. Based on the derived relationship between trajectory segments and the origin, the target is given by:

$$u_{\text{tgt}} = \frac{t}{s}\mathbf{x}_0 - \frac{\mathbf{x}_t(t - s)}{s} - \frac{t^2}{s} \left( 1 - \frac{s}{t} \right) \left( \frac{\partial g_\theta}{\partial t} + \left( \frac{\partial g_\theta}{\partial \mathbf{x}_t} - \mathbf{I} \right)(\mathbf{x}_1 - \mathbf{x}_0) \right). \quad (9)$$

This target avoids complex integral computations while incorporating the ground truth $\mathbf{x}_0$ and the necessary gradient terms to ensure the mapping is consistent with the underlying probability flow ODE. Theoretically, minimizing the distance to this target enforces the model to satisfy the OC condition across arbitrary time intervals.

**Stabilization Factor:** As $s \to 0$, the scaling factor for the target term approaches singularity. We introduce a stabilized weighting term:

$$w(t, s) = \frac{t}{\max(s, \varepsilon)}, \quad (10)$$

where $\varepsilon$ is a numerical stability constant. Furthermore, to prevent extreme gradient variance in the early training stages, we employ $\lambda$ as a warm-up factor to modulate the contribution of the high-order correction terms. The final training target is formulated as:

$$u_{\text{tgt}} = w(t, s)\mathbf{x}_0 - (w(t, s) - 1)\mathbf{x}_t - \lambda w(t, s) (t - s) \left( \frac{\partial g_\theta}{\partial t} + \left( \frac{\partial g_\theta}{\partial \mathbf{x}_t} - \mathbf{I} \right)(\mathbf{x}_1 - \mathbf{x}_0) \right). \quad (11)$$

**Loss Function:** The model parameters $\theta$ are optimized by minimizing the weighted $\ell_2$ distance between the current prediction $u$ and the stop-gradient target $\text{sg}(u_{\text{tgt}})$. Following recent practices in generative modeling (Song et al., 2023; Geng et al., 2025), the sg operator is applied for optimizability by detaching $u_{\text{tgt}}$ from the computation graph, thereby avoiding high gradient computation cost:

$$\mathcal{L}_{\text{OC}}(\theta) = \mathbb{E}_{\mathbf{x}_0, t, s} \left[ \frac{1}{w(t, s)^2} \|u - \text{sg}(u_{\text{tgt}})\|_2^2 \right]. \quad (12)$$

## 3.3. Bridging Graph Space and Latent Space

GOCM adopts a Variational Graph Auto-Encoder (VGAE) (Kipf & Welling, 2016) structure as the latent space encoder, which aims to provide the OC function with a structured prior and an easy-to-sample latent space $\mathbf{z}$.

**Encoder:** Graph Neural Network (GNN) can simultaneously encode node features and graph structure to output independent and identically distributed node embedding vectors. The encoder learns the mean matrix $\boldsymbol{\mu}$ and the log standard deviation $\log \boldsymbol{\sigma}$ of the latent distribution via two dedicated GNNs:

$$\boldsymbol{\mu} = \mathrm{GNN}_{\boldsymbol{\mu}}(\mathcal{G}), \qquad \log \boldsymbol{\sigma} = \mathrm{GNN}_{\boldsymbol{\sigma}}(\mathcal{G}), \quad (13)$$

where $\mathrm{GNN}_{\boldsymbol{\mu}}(\cdot)$ and $\mathrm{GNN}_{\boldsymbol{\sigma}}(\cdot)$ share input network architecture yet separate training processes. Here, we use Graph-SAGE (Hamilton et al., 2017) as GNN. Subsequently, the latent embedding vector is obtained via the reparameterization trick:

$$\mathbf{z} = \boldsymbol{\mu} + \boldsymbol{\sigma}\,\boldsymbol{\eta}, \qquad \boldsymbol{\eta} \sim \mathcal{N}(0, \mathbf{I}). \quad (14)$$

**Decoder:** The decoder maps the latent representation back to the original feature space through a shared decoding head. During training, it is constrained by the reconstruction error, while in the generation phase it is used to decode the attributes of newly synthesized nodes.

**Connection to OC:** The latent representation $\mathbf{z}$ obtained from the VGAE serves as the real data sample $\mathbf{x}_0$ input to the OC model, thereby achieving a seamless integration between the graph structural prior and the generative dynamics.

## 3.4. Heterogeneous Graph Processing Architecture

To enable GOCM to handle heterogeneous graph outlier detection tasks, we design a Multi-input Variational Graph Autoencoder to replace the original VGAE for processing heterogeneous graph data.

Heterogeneous graphs often contain multiple relations $r \in \mathcal{R}$. The MiVGAE captures the heterogeneity of these relations through the following three steps:

**Intra Relation Convolution:** For each relation $r$, its relation-specific GNN convolution is executed to capture the local semantic information:

$$\mathbf{H}^{(r)} = \mathrm{GNN}(\mathcal{G}^{(r)}). \quad (15)$$

**Cross Relation Encoding:** The intermediate representations $\mathbf{H}^{(r)}$ across all relations are concatenated and then merged using a learnable fusion mechanism realized by a neural network to generate a unified node representation $\mathbf{H}$:

$$\mathbf{H} = \phi(\mathbf{W}_{\mathrm{fuse}}\mathrm{CONCAT}(\mathbf{H}^{(1)}, \ldots, \mathbf{H}^{(R)}) + \mathbf{b}_{\mathrm{fuse}}), \quad (16)$$

where $\mathbf{W}_{\mathrm{fuse}}$ and $\mathbf{b}_{\mathrm{fuse}}$ are the learnable fusion parameters, CONCAT denotes the concatenation operation and $\phi$ is an activation function. Finally, the fused node representation $\mathbf{H}$ is projected through two independent linear layers to separately calculate the mean $\boldsymbol{\mu}$ and the log standard deviation $\log \boldsymbol{\sigma}$ of the latent distribution:

$$\boldsymbol{\mu} = \mathbf{W}_{\boldsymbol{\mu}}\mathbf{H} + \mathbf{b}_{\boldsymbol{\mu}}, \quad \log \boldsymbol{\sigma} = \mathbf{W}_{\boldsymbol{\sigma}}\mathbf{H} + \mathbf{b}_{\boldsymbol{\sigma}}. \quad (17)$$

These parameters are then utilized in the subsequent section for interfacing with the OC model. For the detailed decoder architecture used to compute the reconstruction terms, please refer to Appendix C.

## 3.5. Training Objective and Optimization Strategy

GOCM adopts a two-stage optimization strategy to simplify training and ensure efficiency.

### 3.5.1. MiVGAE Stage

**Loss Function of the MiVGAE:** In the first stage, the training of MiVGAE is formulated as a multi-objective optimization problem. To effectively encode the heterogeneous graph, the overall objective function $\mathcal{L}_{\mathrm{total}}$ is defined as a weighted combination of four components, aiming to balance attribute reconstruction, structural learning, relation semantic, and latent space regularization:

$$\mathcal{L}_{\mathrm{total}} = w_x\mathcal{L}_{\mathrm{feat}} + w_e\mathcal{L}_{\mathrm{stru}} + w_p\mathcal{L}_{\mathrm{type}} + \beta\mathcal{L}_{\mathrm{KL}}, \quad (18)$$

where $w_x$, $w_e$, $w_p$, and $\beta$ are hyperparameters that balance the reconstruction term and the Kullback–Leibler (KL) divergence term.

**Node Feature Reconstruction Loss ($\mathcal{L}_{\mathbf{feat}}$):** We minimize the mean squared error between the input features $\mathbf{g}$ and the reconstructed features $\hat{\mathbf{g}}$ in the graph space:

$$\mathcal{L}_{\mathrm{feat}} = \frac{1}{N}\sum_{i=1}^{N}\|\mathbf{g}_i - \hat{\mathbf{g}}_i\|_2^2, \quad (19)$$

where $N$ is the number of nodes in the batch.

**Structure Reconstruction Loss ($\mathcal{L}_{\mathbf{stru}}$):** Link prediction is treated as a binary classification task. For each positive (existing) edge set $\mathcal{E}^+$, we sample an equal-sized set of negative edges $\mathcal{E}^-$. The loss is computed using binary cross-entropy (BCE):

$$\mathcal{L}_{\mathrm{stru}} = -\frac{1}{|E|}\sum_{(u,v)\in E}\Big(y_{uv}\log\phi_s(\hat{s}_{uv}) \\ + (1-y_{uv})\log\big(1-\phi_s(\hat{s}_{uv})\big)\Big), \quad (20)$$

where $E = \mathcal{E}^+ \cup \mathcal{E}^-$, $y_{uv} \in \{0, 1\}$ is the ground-truth edge indicator, $\phi_s(\cdot)$ is the sigmoid function, and $\hat{s}_{uv}$ is the

predicted edge score derived from the latent embeddings $\mathbf{z}_u, \mathbf{z}_v$.

**Edge Type Classification Loss ($\mathcal{L}_{\text{type}}$):** For heterogeneous graphs, we preserve relation semantics by classifying the type of each positive edge. The loss is given by the cross-entropy between the true relation $r_{uv}$ and the predicted type distribution $\hat{\mathbf{p}}_{uv}$:

$$\mathcal{L}_{\text{type}} = -\frac{1}{|\mathcal{E}^+|} \sum_{(u,v)\in\mathcal{E}^+} \sum_{k=1}^{K} \mathbb{I}(r_{uv} = k) \log \hat{p}_{uv}^{(k)}, \quad (21)$$

where $K$ is the total number of relation types and $\mathbb{I}(\cdot)$ is the indicator function.

**KL Divergence Regularization:** We minimize the KL divergence between the variational posterior $q(\mathbf{z} \mid \mathcal{G})$ and $p(\mathbf{z})$:

$$\mathcal{L}_{\text{KL}} = \text{KL}\big(q(\mathbf{z} \mid \mathcal{G}) \,\|\, p(\mathbf{z})\big). \quad (22)$$

The objective of this stage is to enable the latent representation $\mathbf{x}$ to simultaneously capture both node attributes and multi-relational structural information. In the subsequent second stage, following the LDM paradigm (Rombach et al., 2022), we freeze the parameters of MiVGAE and train the consistency mapping of OC exclusively in the latent space.

### 3.5.2. ORIGIN CONSISTENCY STAGE

In the second phase, we optimize the OC function $g_\theta(\mathbf{x}_t, t, s, \mathbf{c})$. The objective is to minimize the OC loss:

$$\mathcal{L}_{\text{OC}} = \mathbb{E}_{\mathbf{x}_0, t, s} \left[ \frac{1}{w(t,s)^2} \left\| u - \text{sg}(u_{\text{tgt}}) \right\|_2^2 \right]. \quad (23)$$

### 3.6. Inference and Generation

The core advantage of GOCM lies in its efficient generation capability, which requires only a single function evaluation (1-NFE). The generation process consists of two main steps: Sample a noise vector from the standard normal distribution, $\hat{\mathbf{x}}_1 \sim \mathcal{N}(0, \mathbf{I})$. Use the trained OC function $g_\theta$ to directly map the noise $\hat{\mathbf{x}}_1$ at $t = 1$ to the data origin at $s = 0$ by one step: $\hat{\mathbf{x}}_0 = g_\theta(\hat{\mathbf{x}}_1, 1, 0, \mathbf{c})$. We treat the synthesized $\hat{\mathbf{x}}$ as the latent representation $\hat{\mathbf{z}}$ (i.e., $\hat{\mathbf{z}} = \hat{\mathbf{x}}_0$). Subsequently, the decoder reconstructs the synthesized nodes from the latent space back into the graph feature space. Furthermore, it determines both the existence and the specific type of edges between node pairs, thereby integrating the synthesized nodes back into the original graph.

## 4. Experiments

This section presents a comprehensive experimental evaluation to validate the effectiveness, efficiency, and generalization capability of our proposed GOCM framework. We first introduce the datasets and baseline models. We then demonstrate the performance of GOCM on various graph outlier detection tasks, followed by ablation studies to verify the importance of our key designs. For a more comprehensive description of the experimental settings, please refer to Appendix D.

### 4.1. Datasets

We use seven different datasets consistent with GADBench: Weibo (Kumar et al., 2019), Elliptic (Weber et al., 2019), Tolokers (Platonov et al., 2023), Questions (Platonov et al., 2023), and DGraph (Huang et al., 2022) are homogeneous graphs, while Amazon (McAuley & Leskovec, 2013) and Yelp (Rayana & Akoglu, 2015) are heterogeneous graphs. These graphs vary significantly in scale, ranging from tens of thousands to millions of nodes. A detailed description of each dataset can be found in Appendix E.

### 4.2. Evaluation Metrics

Following the literature on graph outlier detection, we employ a comprehensive set of metrics to evaluate detector performance. To ensure robustness against class imbalance, we use three key metrics: the Area Under the Receiver Operating Characteristic Curve (AUC), Average Precision (AP), and Recall@k (Rec). Here, $k$ is set to the true number of outliers present in the dataset. Further details regarding these evaluation metrics are provided in Appendix F.

### 4.3. Baseline Models

In our experiments, we evaluate GOCM against baseline methods of different categories. We include three main groups of baselines:

**General Graph Neural Networks:** For homogeneous graphs, we select GCN (Kipf & Welling, 2017), SGC (Wu et al., 2019), GIN (Xu et al., 2019), GraphSAGE (Hamilton et al., 2017), GAT (Veličković et al., 2018), and GT (Shi et al., 2021). For heterogeneous graphs, we adopt HGT (Hu et al., 2020) and RGCN (Schlichtkrull et al., 2018) as provided in GADBench.

**Graph Outlier Detectors:** In addition to general GNNs, we evaluate eleven dedicated graph outlier detectors: GAS (Li et al., 2019), DCI (Wang et al., 2021), PCGNN (Liu et al., 2021), BernNet (He et al., 2021), GATSep (Zhu et al., 2020), AMNet (Chai et al., 2022), BWGNN (Tang et al., 2022), CARE-GNN (Dou et al., 2020a), H2-FDetector (Shi et al., 2022), GHRN (Gao et al., 2023) and SpaceGNN (Dong et al., 2025).

**Graph Outlier Data Augmentation Methods:** Since data augmentation for graph outlier detection is a relatively new research direction, available baselines are limited. We com-

*Table 1.* Performance in AUC, AP, and Rec (%) on four datasets.

| DATASET | WEIBO | | | TOLOKERS | | | QUESTIONS | | | ELLIPTIC | | |
|---------|-------|----|-----|----------|----|-----|-----------|----|-----|----------|----|-----|
| METRIC | AUC | AP | REC | AUC | AP | REC | AUC | AP | REC | AUC | AP | REC |
| GCN | 98.11 | 93.48 | 89.34 | 74.69 | 42.88 | 42.06 | 69.81 | 12.54 | 16.99 | 82.68 | 22.23 | 27.61 |
| SGC | 98.66 | 92.46 | 87.90 | 70.67 | 38.03 | 35.98 | 69.88 | 10.13 | 15.62 | 73.02 | 11.44 | 9.14 |
| GIN | 97.47 | 92.67 | 87.90 | 74.05 | 36.57 | 36.76 | 67.76 | 12.30 | 18.36 | 84.38 | 29.66 | 35.64 |
| GRAPHSAGE | 96.54 | 89.25 | 86.17 | 79.42 | 48.65 | 46.42 | 71.69 | 17.63 | 21.10 | 85.31 | 37.52 | 36.20 |
| GAT | 94.08 | 90.25 | 86.74 | 77.26 | 43.14 | 43.30 | 70.33 | 14.51 | 17.26 | 84.42 | 23.43 | 27.42 |
| GT | 97.06 | 91.44 | 87.03 | 79.24 | 46.22 | 46.57 | 70.83 | 16.14 | 20.27 | 87.14 | 29.91 | 38.97 |
| GAS | 94.88 | 90.70 | 86.74 | 76.91 | 47.35 | 45.02 | 64.50 | 13.61 | 17.53 | 87.81 | 40.03 | 44.78 |
| DCI | 93.90 | 87.78 | 83.86 | 75.98 | 39.85 | 40.19 | 67.95 | 14.58 | 19.18 | 81.93 | 27.63 | 33.15 |
| PCGGN | 90.89 | 84.57 | 79.83 | 72.18 | 37.52 | 36.76 | 68.38 | 14.79 | 16.99 | 86.50 | 42.66 | 43.77 |
| GATSEP | 96.72 | 91.55 | 89.05 | 79.63 | 46.08 | 46.73 | 69.96 | 15.98 | 19.18 | 83.89 | 21.46 | 21.15 |
| BERNNET | 93.85 | 88.00 | 85.30 | 76.20 | 42.20 | 42.21 | 70.80 | 16.04 | 17.53 | 82.01 | 20.52 | 23.55 |
| AMNET | 95.88 | 89.74 | 85.59 | 75.83 | 42.66 | 41.90 | 69.71 | 17.02 | 19.18 | 80.06 | 16.73 | 17.17 |
| BWGNN | 98.29 | 92.72 | 84.73 | 80.15 | 49.65 | 47.35 | 69.47 | 16.24 | 18.63 | 84.32 | 22.56 | 26.50 |
| GHRN | 97.21 | 92.67 | 88.18 | 79.80 | 49.50 | 48.29 | 68.24 | 16.24 | 18.63 | 85.36 | 24.01 | 30.29 |
| SPACEGNN | 96.40 | 86.40 | 79.50 | 71.50 | 36.20 | 36.60 | 65.00 | 9.70 | 14.50 | — | — | — |
| DAGAD | 98.54 | 83.36 | 90.78 | 77.69 | 33.94 | 44.39 | 71.21 | 6.88 | 20.55 | 85.62 | 26.18 | 40.54 |
| GODM | 99.57 | 97.54 | 93.08 | 83.46 | 52.95 | 52.96 | 76.84 | **20.48** | **24.66** | **89.77** | 43.92 | 53.92 |
| **GOCM** | **99.62** | **97.92** | **93.94** | **83.69** | **55.39** | **53.86** | **77.08** | 19.39 | 24.04 | 88.40 | **46.02** | **54.01** |

pare with DAGAD (Liu et al., 2022) and GODM (Liu et al., 2025), where DAGAD is designed specifically for homogeneous graphs. Detailed descriptions of each baseline are provided in Appendix G.

## 4.4. Main Results

This section presents the performance comparison between GOCM and all baseline models, evaluated separately on homogeneous and heterogeneous graphs.

### 4.4.1. PERFORMANCE ON HOMOGENEOUS GRAPHS

We first evaluate the effectiveness of GOCM on graph outlier detection tasks on homogeneous datasets. Table 1 reports the AUC, AP, and Rec performance of various algorithms on the Weibo, Elliptic, Tolokers, and Questions datasets. For each metric, the best score is highlighted in bold and the second-best is underlined.

Our observations are as follows. Among the baselines, GODM demonstrates a significant improvement in graph outlier detection performance across all datasets and metrics. Furthermore, our GOCM comprehensively outperforms GODM across all three key metrics: AUC, AP, and Rec, on the Weibo and Tolokers datasets. Notably, on the Tolokers dataset, GOCM achieves a 2.44% improvement in AP. This result suggests that the outliers synthesized by GOCM more accurately cover the latent anomaly distribution, thereby enabling the model to effectively reduce false positives. It is noteworthy that on the Weibo dataset, where GODM already attains a high AUC of 99.57, GOCM further improves the performance, achieving an AUC of 99.62.

Consistent improvements are also observed on the DGraph dataset, as detailed in Appendix H.

### 4.4.2. PERFORMANCE ON HETEROGENEOUS GRAPHS

*Table 2.* Performance in AUC, AP, and Rec (%) on Heterogeneous Graphs: Amazon and Yelp.

| DATASET | YELP | | | AMAZON | | |
|---------|------|----|-----|--------|----|-----|
| METRIC | AUC | AP | REC | AUC | AP | REC |
| RGCN | 78.34 | 34.57 | 34.98 | 92.03 | 67.97 | 65.49 |
| HGT | 89.62 | 62.63 | 57.75 | 89.64 | 71.46 | 70.22 |
| H2DETECTOR | 89.07 | 59.40 | 57.54 | 78.66 | 39.95 | 44.35 |
| CARE-GNN | 90.85 | 69.47 | 65.90 | 90.84 | 72.64 | 67.72 |
| GODM | 89.41 | 64.64 | 62.00 | 92.35 | 69.30 | 66.84 |
| **GOCM** | 92.05 | 72.15 | 68.04 | 93.36 | 72.13 | 69.72 |

In Table 2, we benchmark the performance of heterogeneous graph outlier detectors on the Yelp and Amazon datasets, which contain multiple types of edges (relations). As shown in the table, GOCM outperforms the heterogeneous graph detectors provided in GADBench and also surpasses the performance enhanced by GODM. This demonstrates the strong competitiveness and effectiveness of our GOCM framework on heterogeneous graphs.

### 4.4.3. STUDY ON ENHANCEMENT EFFECTS FOR DOWNSTREAM MODELS

We further investigate whether GOCM consistently enhances downstream models. Taking the Tolokers dataset as an example, we integrate GOCM with all compared graph outlier detection algorithms and report the performance changes in Table 3 across three metrics: AUC, AP, and

*Table 3.* Performance improvements in AUC, AP, and Rec (%) on Tolokers datasets.

| METRIC | BY GODM | | | BY GOCM | | |
|---|---|---|---|---|---|---|
| | AUC | AP | REC | AUC | AP | REC |
| GCN | 75.45 (+0.76) | 44.17 (+1.29) | 44.24 (+2.18) | 79.00 (+4.31) | 49.40 (+6.52) | 47.85 (+5.57) |
| SGC | 72.73 (+2.06) | 39.75 (+1.72) | 38.01 (+2.02) | 72.50 (+1.83) | 39.85 (+1.82) | 38.09 (+2.11) |
| GIN | 74.83 (+0.78) | 38.54 (+1.96) | 38.32 (+1.56) | 75.81 (+1.76) | 40.37 (+3.80) | 39.37 (+2.61) |
| GRAPHSAGE | 81.65 (+2.23) | 52.53 (+3.87) | 50.93 (+4.52) | 81.75 (+2.33) | 52.59 (+3.94) | 50.04 (+3.62) |
| GAT | 82.18 (+4.91) | 51.13 (+7.99) | 50.78 (+7.48) | 82.44 (+5.18) | 52.54 (+9.40) | 50.81 (+7.51) |
| GT | 82.73 (+3.49) | 51.73 (+5.51) | 50.93 (+4.36) | 82.84 (+3.60) | 53.41 (+7.19) | 52.47 (+5.90) |
| GAS | 76.96 (+0.05) | 45.48 (−1.87) | 43.15 (−1.87) | 77.67 (+0.27) | 46.08 (+0.38) | 44.68 (−0.32) |
| DCI | 73.75 (−2.23) | 37.52 (−2.33) | 37.54 (−2.65) | 76.35 (+0.37) | 41.45 (+1.60) | 40.59 (+0.40) |
| PCGGN | 73.65 (+1.47) | 38.42 (+0.90) | 38.01 (+1.25) | 73.30 (+1.12) | 38.82 (+1.30) | 38.23 (+1.47) |
| GATSEP | 83.46 (+3.83) | 52.95 (+6.87) | 52.80 (+6.07) | 83.69 (+4.06) | 55.39 (+9.31) | 53.86 (+7.13) |
| BERNNET | 76.53 (+0.33) | 44.04 (+1.83) | 42.21 (+0.00) | 77.64 (+1.44) | 45.27 (+3.07) | 43.87 (+1.66) |
| AMNET | 76.67 (+0.84) | 44.31 (+1.65) | 43.30 (+1.40) | 77.78 (+1.95) | 45.36 (+2.70) | 44.15 (+2.26) |
| BWGNN | 82.10 (+1.95) | 51.94 (+2.29) | 51.25 (+3.89) | 82.26 (+2.11) | 53.06 (+3.41) | 51.66 (+4.31) |
| GHRN | 82.05 (+2.25) | 51.84 (+2.34) | 51.56 (+3.27) | 82.65 (+2.85) | 53.48 (+3.98) | 52.13 (+3.84) |

Rec. The absolute performance values are shown outside the parentheses, while the relative changes are indicated inside.

It can be observed that while GODM improves performance for most algorithms, it leads to degradation for two detectors, DCI and GAS. Our GOCM mitigates this issue: only GAS exhibits a minor drop in the Rec metric, whereas GOCM achieves better results on nearly all other detectors. For instance, it improves the AP metric of GAT by 9.40 and that of GATSep by 9.31.

### 4.5. Ablation Study

To validate the contribution of OC in generating high-fidelity outlier samples, we conduct an ablation study on the Weibo and Tolokers datasets. As shown in Table 4, the OC mechanism consistently achieves better detection performance compared to the standard Consistency Model across both datasets. These results suggest that our interval-based origin inference strategy provides precise guidance for effective single-step synthesis. For more comprehensive evaluation across more detectors and datasets, as well as the ablation analysis of components, please refer to Appendix I.

*Table 4.* Ablation study on the effectiveness of different components (CM vs. OC) on Tolokers and Weibo datasets (AUC, AP, and Rec in %).

| DATASET | GCN | GATSEP | CM | OC | AUC | AP | REC |
|---|---|---|---|---|---|---|---|
| WEIBO | √ | | | | 98.11 | 93.48 | 89.34 |
| | | √ | | | 96.72 | 91.55 | 89.05 |
| | √ | | √ | | 99.52 | 97.43 | 93.25 |
| | √ | | | √ | 99.62 | 97.92 | 93.94 |
| TOLOKERS | √ | | | | 74.69 | 42.88 | 42.06 |
| | | √ | | | 79.63 | 46.08 | 46.73 |
| | | √ | √ | | 82.73 | 53.55 | 52.95 |
| | | √ | | √ | 83.69 | 55.39 | 53.86 |

### 4.6. Efficiency Comparison

We perform rigorous benchmarking on the generation speed of GOCM and compare it with the iterative model GODM. Fig. 3 records the time cost required to generate the same number of synthetic samples, verifying the efficiency advantage of the OC mechanism in single-step generation. Across all datasets, the generation time of OC is significantly lower than that of the diffusion model used in GODM. It is noted that on the Tolokers dataset, which has the smallest outlier node feature dimension and the highest proportion of outlier nodes among the four datasets, our single-step generation does not follow the same time-variation trend as GODM, yet still reduces the time cost by an order of magnitude.

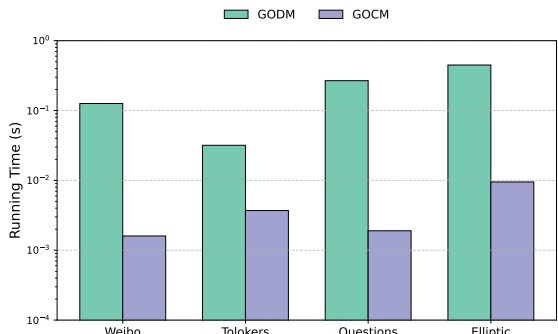

*Figure 3.* **Generation Time Comparison.** This figure compares the generation time of the diffusion model and the OC model.

In practical applications, decoupling the pre-training or augmentation phase from downstream tasks is a common strategy to balance model performance and computational overhead (Zhou et al., 2023). Specifically, GOCM functions as an independent data augmentation plug-in. Once the generative model undergoes its training and produces the synthetic outliers, this augmented data can be reused across

various downstream GNN detectors. In contrast, joint training methods require re-executing the entire pipeline when the downstream detection architecture is updated.

To further demonstrate our efficiency advantage within the family of generative methods, we compare the wall-clock time of GOCM with diffusion-based models (GODM and DiffGAD (Li et al., 2025)). As shown in Table 5, GOCM consistently requires less time than multi-step diffusion baselines.

*Table 5.* The wall-clock time (s) comparison includes both the training and inference stages and OOM denotes out of memory.

| METHOD | WEIBO | QUESTIONS | DGRAPH |
|---|---|---|---|
| | TIME | TIME | TIME |
| DIFFGAD | 135.24 | OOM | OOM |
| GODM | 33.01 | 172.35 | 683.95 |
| **GOCM** | **23.48** | **71.13** | **413.68** |

### 4.7. Sensitivity Analysis

To further evaluate the robustness of our proposed framework, we conduct a sensitivity analysis on the key hyperparameters introduced in GOCM. It is crucial to clarify that the stabilization weight $w(t, s)$ is a mathematically derived adaptive factor rather than a heuristically tuned variable.

Therefore, our sensitivity analysis primarily focuses on the warm-up factor $\lambda$ (denoted as $\lambda_{base}$ in Appendix B), which modulates the gradient correction terms during the early training stages. We vary the value of $\lambda$ from -1.0 to 0.5 and evaluate its impact on the downstream anomaly detection performance using the Weibo dataset.

As illustrated in Fig. 4, the mean performance metrics (AUC, AP, and Rec) remain stable across various assigned values of $\lambda$. These results indicate that GOCM maintains architectural stability and generates outliers without the need for extensive hyperparameter tuning.

### 4.8. Additional Experimental Analysis

We provide additional experimental analysis of the proposed GOCM framework in Appendix H, such as the visualization of feature density, robustness analysis across different training stages, comparisons of edge generation mechanisms and evaluation of model robustness under extreme label scarcity.

## 5. Conclusion

In this work, we proposed GOCM, a novel data augmentation framework for supervised graph outlier detection. GOCM is designed to address the pervasive and critical challenge of class imbalance in this domain. Notably, GOCM is model-agnostic and can be flexibly integrated with various downstream graph outlier detectors. Graph outlier detec-

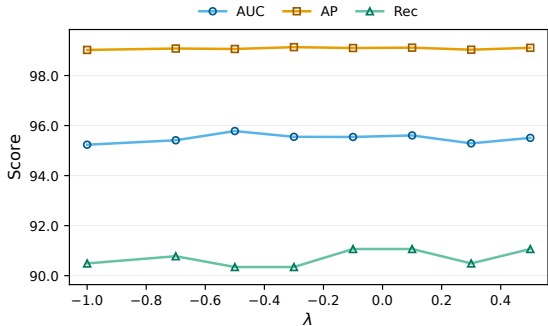

*Figure 4.* **Sensitivity Analysis of the Warm-up Factor.** Evaluation of downstream detection performance (AUC, AP, and Rec in %) on the Weibo dataset across different values of the warm-up factor $\lambda$.

tion remains a challenging research area. In this work, we chose to perform generation in the latent space to facilitate heterogeneous encoding and reduce computational costs. Future work could explore the direct application and adaptation of diffusion models within the original graph space. Furthermore, while our generative model is built upon the consistency model, other promising frameworks such as Rectified Flow, which is grounded in velocity field theory, have demonstrated remarkable results. Extending and applying such advanced generative models to graph synthesis presents a valuable direction for future research.

## Acknowledgements

This work is supported in part by the Qingdao Natural Science Foundation under Grant 24-4-4-zrjj-126-jch, in part by the Guangxi Science and Technology Program under Grant FN2504240022, and in part by the National Natural Science Foundation of China under Grant 62502287.

## Impact Statement

Our work is committed to addressing the long-standing challenge of class imbalance in graph-based learning by proposing GOCM, a highly efficient generative modeling framework. In critical sectors such as financial fraud detection and anti-money laundering , where the extreme scarcity of anomalies often limits the generalization power of detection models , the framework provided by GOCM can help downstream models enhance their detection capabilities. While this framework is designed to empower defensive systems, it is necessary to acknowledge that high-fidelity graph synthesis technology theoretically possesses potential dual-use risks, such as being used to simulate deceptive patterns. Consequently, we emphasize the importance of adhering to responsible research guidelines to ensure its application remains aligned with the principles of safeguarding

cyberspace and social integrity.

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

## A. Detailed Related Work

Graph Outlier Detection has emerged as a highly popular and critical research direction in the field of graph machine learning (Tang et al., 2023). This technique focuses on identifying nodes or subgraphs that significantly deviate from predominant normal patterns, and has demonstrated its indispensable value in high-risk domains such as fraud detection (Huang et al., 2022; Pan et al., 2025), spam detection (Dou et al., 2020b), anti-money laundering (Weber et al., 2019), and abnormal pattern detection in social networks (Zhang et al., 2025). Various methods have achieved significant success, including those based on structural anomalies or attribute deviations (Ding et al., 2019; Fan et al., 2020), and those utilizing latent geometric properties of graphs such as curvature for anomaly detection (Grover et al., 2025). However, despite these significant advances, the field of graph outlier detection has persistently been plagued by a fundamental challenge: class imbalance, where normal nodes vastly outnumber anomalous nodes (Tang et al., 2022). For instance, in the DGraph dataset (Huang et al., 2022), which simulates a real-world financial network, outliers constitute a mere 1.3%. The presence of such extreme class imbalance severely compromises the training of robust detectors. This is because the loss function exhibits a pronounced bias toward the majority negative class (Liu et al., 2025), consequently yielding models with limited generalization power and suboptimal identification of actual outliers.

To address this challenge, data-level solutions such as upsampling and downsampling (e.g., in (Dou et al., 2020a)) and feature-level methods that generate new representations by shuffling and splicing low-dimensional node embeddings (Liu et al., 2022) have been employed. However, traditional interpolation-based methods often fail to adequately capture the complex underlying distribution of graph data, resulting in limited performance. This underscores the pressing need for more advanced data augmentation paradigms.

In recent years, generative models have achieved remarkable success in the field of data synthesis. Among them, Diffusion Models (Ho et al., 2020) have set a new benchmark for high-quality sample generation by performing stepwise denoising from a simple prior distribution. In the latest research, some work has leveraged the Latent Diffusion Model (LDM) (Rombach et al., 2022) to introduce controlled perturbations into the latent space of graphs for generating pseudo-anomalous graphs (Cai et al., 2025). In parallel, another line of work has leveraged LDM to directly achieve effective synthesis of graph outliers. The synthesized graphs outliers can be combined with real graphs to train downstream graph anomaly detectors (Liu et al., 2025).

## B. Derivation of Origin Consistency Model

### B.1. Trajectory Dynamics and Consistency Mapping

To establish a mapping from any point $\mathbf{x}_t$ to the origin, we analyze the linear trajectory dynamics of the probability Flow. Given the linear interpolation $\mathbf{x}_u = \mathbf{x}_0 + u(\mathbf{x}_1 - \mathbf{x}_0)$, the displacement from the origin is proportional to time $u$. Consequently, the instantaneous velocity is given by $\mathbf{v} = (\mathbf{x}_u - \mathbf{x}_0)/u$. Leveraging this property, we formulate our consistency function $g(\mathbf{x}_t, t, s)$ by integrating the estimated velocity field. Instead of directly modeling the abstract score function, we explicitly utilize the time-normalized drift term:

$$g(\mathbf{x}_t, t, s) = \mathbf{x}_t + \frac{t}{t-s} \int_t^s \frac{\mathbf{x}_u - \mathbb{E}[\mathbf{x} \mid \mathbf{x}_u]}{u} \, du \tag{24}$$

Here, the term $\frac{\mathbf{x}_u - \mathbb{E}[\mathbf{x}|\mathbf{x}_u]}{u}$ represents the velocity estimate at time $u$. Integrating this velocity yields the robust trajectory evolution. Although this integral form shares structural similarities with the parameterization used in CTM (Kim et al., 2024), our derivation is grounded in the geometry of linear probability flows rather than standard diffusion dynamics.

We consider that $g(\mathbf{x}_t, t, s)$ embodies the modeling philosophy of "Interval-based Origin Inference": specifically, by analyzing the dynamic evolution of the noisy state $\mathbf{x}_t$ within a time interval, the model can directly invert the corresponding initial state. In a physical sense, $g$ encapsulates the trajectory flow information from $t$ to $s$.

### B.2. Trajectory Derivative Derivation of $g(\mathbf{x}_t, t, s)$

To train the model $g_\theta$, we need to establish its mathematical relationship with the ideal denoiser $\mathbb{E}[\mathbf{x} \mid \mathbf{x}_t]$. We start by rearranging the definition of $g$:

$$\frac{t-s}{t}\big(g(\mathbf{x}_t, t, s) - \mathbf{x}_t\big) = \int_t^s \frac{\mathbf{x}_u - \mathbb{E}[\mathbf{x} \mid \mathbf{x}_u]}{u} \, du \tag{25}$$

We then differentiate both sides with respect to time $t$.

**Left-side Derivative:**

$$\frac{\partial}{\partial t}(\text{Left}) = \frac{s}{t^2}(g(\mathbf{x}_t, t, s) - \mathbf{x}_t) + \frac{t-s}{t}\left(\frac{\partial g(\mathbf{x}_t, t, s)}{\partial t} + \left(\frac{\partial g(\mathbf{x}_t, t, s)}{\partial \mathbf{x}_t} - \mathbf{I}\right)\frac{\partial \mathbf{x}_t}{\partial t}\right) \tag{26}$$

**Right-side Derivative:**

$$\frac{d}{dt}(\text{Right}) = -\left(\frac{\mathbf{x}_t - \mathbb{E}[\mathbf{x} \mid \mathbf{x}_t]}{t}\right) \tag{27}$$

By equating the two sides, we obtain the differential constraint equation describing the trajectory consistency:

$$\frac{s}{t^2}(g(\mathbf{x}_t, t, s) - \mathbf{x}_t) + \frac{t-s}{t}\left(\frac{\partial g(\mathbf{x}_t, t, s)}{\partial t} + \left(\frac{\partial g(\mathbf{x}_t, t, s)}{\partial \mathbf{x}_t} - \mathbf{I}\right)\frac{\partial \mathbf{x}_t}{\partial t}\right) = -\frac{\mathbf{x}_t - \mathbb{E}[\mathbf{x} \mid \mathbf{x}_t]}{t} \tag{28}$$

### B.3. Solving for the Ideal Denoiser $\mathbb{E}[\mathbf{x} \mid \mathbf{x}_t]$

By solving the equation above, we can derive the expression for the ideal denoiser:

$$\mathbb{E}[\mathbf{x} \mid \mathbf{x}_t] = \frac{s}{t}(g(\mathbf{x}_t, t, s) - \mathbf{x}_t) + (t-s)\left(\frac{\partial g(\mathbf{x}_t, t, s)}{\partial t} + \left(\frac{\partial g(\mathbf{x}_t, t, s)}{\partial \mathbf{x}_t} - \mathbf{I}\right)\frac{\partial \mathbf{x}_t}{\partial t}\right) + \mathbf{x}_t \tag{29}$$

### B.4. Neural Denoiser $f_\theta$ and Gradient Correction

We substitute the parameterized model $g_\theta$ to construct the parameterized ideal denoiser $f_\theta$. To capture the trajectory characteristics within the graph latent space, we approximate the time derivative $\frac{\partial \mathbf{x}_t}{\partial t}$ using the instantaneous velocity $(\mathbf{x}_1 - \mathbf{x}_0)$ derived from the linear interpolation noise model (following the ReFlow framework):

$$f_\theta(\mathbf{x}_t, t, s) = \mathbf{x}_t + \frac{s}{t}(g_\theta - \mathbf{x}_t) + (t-s)\left(\frac{\partial g_\theta}{\partial t} + \left(\frac{\partial g_\theta}{\partial \mathbf{x}_t} - \mathbf{I}\right)(\mathbf{x}_1 - \mathbf{x}_0)\right) \tag{30}$$

Ideally, this objective should contain the derivatives of the true function $g$ (i.e., $\partial g$); however, in practice, these are replaced by their parameterized counterparts (i.e., $\partial g_\theta$). Consequently, we naturally substitute $f_\theta$ into the fundamental consistency loss function:

$$\mathcal{L}_{\text{CM}}(\theta) = \mathbb{E}\left[\|f_\theta(\mathbf{x}_t, t, s) - \mathbf{x}_0\|^2\right] \tag{31}$$

By substituting the expression for $f_\theta(\mathbf{x}_t, t, s)$, we derive a valid objective function for training $g_\theta(\mathbf{x}_t, t, s)$:

$$\mathcal{L}(\theta) = \mathbb{E}_{t\sim\mathbb{U}(0,1), \mathbf{x}_0\sim p(\mathbf{x}_0), \mathbf{x}_t\sim q(\mathbf{x}_t|\mathbf{x}_0)}\left[\left\|\mathbf{x}_t + \frac{s}{t}\left(g_\theta(\mathbf{x}_t, t, s) - \mathbf{x}_t\right) + (t-s)\left(\frac{\partial g_\theta}{\partial t} + \left(\frac{\partial g_\theta}{\partial \mathbf{x}_t} - \mathbf{I}\right)(\mathbf{x}_1 - \mathbf{x}_0)\right) - \mathbf{x}_0\right\|^2\right] \tag{32}$$

To facilitate more efficient and robust training, we rearrange the optimization objective as follows:

$$\mathcal{L}(\theta) = \mathbb{E}\left[\left(\frac{s}{t}\right)^2\left\|g_\theta + \left(\frac{\mathbf{x}_t(t-s)}{s} + \frac{t^2}{s}\left(1 - \frac{s}{t}\right)\left(\frac{\partial g_\theta}{\partial t} + \left(\frac{\partial g_\theta}{\partial \mathbf{x}_t} - \mathbf{I}\right)(\mathbf{x}_1 - \mathbf{x}_0)\right)\right) - \frac{t}{s}\mathbf{x}_0\right\|^2\right] \tag{33}$$

This yields a theoretically ideal outcome: a single, explicit minimization objective. These properties imply that, regardless of the optimization algorithm employed, finding the minimum of the expression above guarantees the acquisition of the desired model, specifically a generative model that is theoretically capable of single-step generation.

### B.5. Construction of Regression Target and Numerical Stabilization Strategies

The computational cost of the aforementioned loss function is relatively high as it inherently necessitates second-order gradients. Consequently, we introduce a regression target $\mathbf{u}_{\text{tgt}}$:

$$\mathcal{L}(\theta) = \mathbb{E}\left[\left(\frac{s}{t}\right)^2 \cdot \left\|g_\theta(\mathbf{x}_t, t, s) - \text{stopgrad}(\mathbf{u}_{\text{tgt}})\right\|^2\right] \tag{34}$$

The target $\mathbf{u}_{\text{tgt}}$ is defined to include the gradient correction term $\mathcal{J}$:

$$\mathbf{u}_{\text{tgt}} = \frac{t}{s}\mathbf{x}_0 - \frac{t-s}{s}\mathbf{x}_t - \frac{t^2}{s}\left(1 - \frac{s}{t}\right) \cdot \mathcal{J} \tag{35}$$

where $\mathcal{J}$ is given by:

$$\mathcal{J} = \frac{\partial g_\theta}{\partial t} + \left(\frac{\partial g_\theta}{\partial \mathbf{x}_t} - \mathbf{I}\right)(\mathbf{x}_1 - \mathbf{x}_0) \tag{36}$$

In the loss function, following common practices (e.g., MeanFlow), we apply a stop-gradient (sg) operation to the target $\mathbf{u}_{\text{tgt}}$. In our method, this operation avoids "double backpropagation" caused by the Jacobian-vector product (JVP), thereby circumventing the overhead of high-order optimization.

### B.6. Stabilization Factors

In the context of single-step prediction tasks, the weighting term $t/s$ leads to numerical divergence as $s \to 0$. To mitigate this, we introduce a stabilization factor:

$$w(t, s) = \frac{t}{\max(s, \varepsilon)} \tag{37}$$

where $\varepsilon$ represents a minimal numerical constant. By imposing a bound at extremely small time scales, this factor effectively suppresses extreme numerical fluctuations during the early stages of training. Furthermore, we observed that the high-order gradient correction term can cause instability during the initial training phase. To alleviate this issue and enhance training efficiency, we introduce an additional modulation factor $\lambda(k)$. Let $k$ denote the current training step and $K_{\text{warm}}$ denote the total warm-up steps. The modulation factor is defined as:

$$\lambda(k) = \lambda_{\text{base}} \cdot \min\left(1, \frac{k}{K_{\text{warm}}}\right). \tag{38}$$

Here, $\lambda_{\text{base}}$ is a tunable hyperparameter controlling the final weight of the correction term. This design encourages the model to prioritize learning simple consistency mappings—effectively ignoring complex gradient correction terms at the start of training ($k < K_{\text{warm}}$,), and subsequently introduces high-order gradient information gradually as the model stabilizes.

$$\mathbf{u}_{\text{tgt}} = w(t,s)\mathbf{x}_0 - (w(t,s) - 1)\mathbf{x}_t - \lambda(k)w(t,s)(t-s)\mathcal{J}(\mathbf{x}_t, t, s) \tag{39}$$

Through this approach, GOCM is capable of directly perceiving the trajectory flow from noise, achieving a conceptual leap from empirical constraints to dynamical inference. This ensures high sample fidelity within the single-step generation paradigm.

### B.7. Applicability of the Linear Trajectory to Non-linear Graph Flows

Regarding the linear trajectory and its applicability to complex non-linear graph flows, we emphasize that according to Reflow theory (Liu et al., 2023), any complex non-linear target distribution can be exactly reconstructed by marginalizing linear conditional paths. The linear trajectory only defines the "straight-line displacement" of individual sample pairs; the aggregated global probability flow still fits complex manifolds.

### B.8. Pseudocode for GOCM Training and Generation

We present the pseudocode for both the training and single-step generation phases of our Origin Consistency Model in Algorithm 1 and Algorithm 2, respectively.

---

**Algorithm 1** Training

---

```
# fn(z, t, s): Origin Consistency Neural Network
# x: latent batch from MiVGAE encoder (z in VAE context)

# Sample time-steps and standard Gaussian noise
t, s = sample_t_s(batch_size)
e = randn_like(x)

z = (1 - t) * x + t * e

u = fn(z, t, s)

# dg_dt = ∂fn/∂t, jvp_x = (∂fn/∂z) · (e - x)
dg_dt, jvp_x = compute_gradients(fn, z, t, s, direction=(e - x))

grad_term = dg_dt + jvp_x - (e - x)
u_tgt = calc_target(x, z, t, s, grad_term)

loss = metric(u - stopgrad(u_tgt))
```

---

**Algorithm 2** Inference

---

```
e = randn(x_shape)
x_syn = fn(e, t=1, s=0)
```

---

# C. Detailed Architecture of MiVGAE

We provide the detailed mathematical formulation of the MiVGAE decoder. The decoder is responsible for reconstructing node attributes, identifying edge existence, and classifying edge types from the latent space.

Let $\mathbf{z} \in \mathbb{R}^{N \times d}$ denote the latent representation output by the encoder (corresponding to $\mathbf{x}_0$), where $N$ is the number of nodes and $d$ is the latent dimension. Let $\mathbf{y}$ represent the node labels and $K$ represent the number of relation types.

### C.1. Label Conditioning

Before decoding, we inject label information into the latent representation to guide the generation process. This is achieved via a learnable linear projection:

$$\tilde{\mathbf{z}} = \mathbf{z} + \mathbf{W}_{\text{label}}\,\mathbf{y}, \tag{40}$$

where $\mathbf{W}_{\text{label}}$ is the weight matrix mapping the label space to the latent dimension. This conditioned representation $\tilde{\mathbf{z}}$ is utilized for all subsequent decoding tasks.

### C.2. Node Attribute Decoder

The node attribute decoder reconstructs the original feature matrix $\hat{\mathcal{G}}$. For a specific node $\mathbf{z}_u$, the reconstruction $\hat{\mathbf{g}}_u$ is computed as:

$$\hat{\mathbf{g}}_u = \mathbf{W}_{\text{attr}}\tilde{\mathbf{z}}_u + \mathbf{b}_{\text{attr}}, \tag{41}$$

where $\mathbf{W}_{\text{attr}}$ and $\mathbf{b}_{\text{attr}}$ correspond to the weights and bias of the linear decoding layer.

### C.3. Edge Structure Decoder

To predict the existence of an edge between two nodes $u$ and $v$, we first concatenate their conditioned latent representations to form a pair-wise embedding $\mathbf{h}_{uv} = \text{CONCAT}(\tilde{\mathbf{z}}_u, \tilde{\mathbf{z}}_v)$, where CONCAT denotes the concatenation operation. The edge existence score $\hat{s}_{uv}$ is then computed as:

$$\hat{s}_{uv} = \mathbf{W}_{\text{stru}}\mathbf{h}_{uv} + b_{\text{stru}}, \tag{42}$$

where $\mathbf{W}_{\text{stru}} \in \mathbb{R}^{1 \times 2d}$ and $b_{\text{stru}} \in \mathbb{R}$ are the learnable parameters. This score corresponds to the input of the sigmoid function $\phi_s(\cdot)$ in Eq. (20).

### C.4. Edge Type Decoder

We utilize a dedicated decoding head that maps the pair-wise embedding $\mathbf{h}_{uv}$ to a vector $\mathbf{p}_{uv}$:

$$\mathbf{p}_{uv} = \mathbf{W}_{\text{type}}\mathbf{h}_{uv} + \mathbf{b}_{\text{type}}, \tag{43}$$

where $\mathbf{W}_{\text{type}}$ and $\mathbf{b}_{\text{type}}$ are the learnable parameters. Each element of $\mathbf{p}_{uv}$ is used to compute $\mathcal{L}_{\text{type}}$ in Eq. (21).

## D. Experimental Setting

To ensure the reproducibility and fairness of our comparative study, we adhere to the following experimental protocols.

For all baseline methods, including general Graph Neural Networks (e.g., GCN, GAT) and specialized Graph Outlier Detectors (e.g., BWGNN, GHRN), we strictly follow the default hyperparameter configurations and implementation standards provided by the GADBench. This ensures that our comparisons are made against standard, optimized versions of these models.

To mitigate the influence of random initialization, all experimental results reported in this paper (including AUC, AP, and Rec) are calculated as the mean values derived from multiple independent runs with different random seeds.

For the generative augmentation method, GODM, we configure the diffusion reverse process with the default setting of 50 sampling steps. For our proposed GOCM framework, by default, the number of generated synthetic nodes is set to match the number of labeled anomaly nodes in the training set.

All experiments are conducted on a single NVIDIA GeForce RTX 3090 GPU. The specific hyperparameters for our proposed GOCM framework and the software environment used for these experiments are comprehensively documented in the accompanying source code.

## E. Detailed Dataset Descriptions

In our experimental evaluation, we utilize seven benchmark datasets provided by GADBench, covering diverse domains including social networks, e-commerce, crowdsourcing platforms, Q&A communities, and financial transactions. The detailed statistics and construction descriptions for each dataset are provided below:

*Table 6.* Summary of experimental datasets, including node counts, edge counts, anomaly ratio, training ratio, and feature types (Misc. signifies mixed attributes).

| NAME | #NODES | #EDGES | ANOMALY | TRAIN | FEATURE TYPE |
|---|---|---|---|---|---|
| WEIBO | 8,405 | 407,963 | 10.3% | 40% | TEXT EMBEDDING. |
| TOLOKERS | 11,758 | 519,000 | 21.8% | 40% | MISC. INFORMATION. |
| QUESTIONS | 48,921 | 153,540 | 3.0% | 52% | TEXT EMBEDDING. |
| ELLIPTIC | 203,769 | 234,355 | 9.8% | 50% | MISC. INFORMATION. |
| DGRAPH | 3,700,550 | 4,300,999 | 1.3% | 70% | MISC. INFORMATION. |
| AMAZON (HETERO) | 11,944 | 4,398,392 | 9.5% | 70% | MISC. INFORMATION. |
| YELP (HETERO) | 45,954 | 3,846,979 | 14.5% | 70% | MISC. INFORMATION. |

### E.1. Weibo

Derived from the Tencent Weibo platform (Kumar et al., 2019), this dataset constructs a User-Topic graph focusing on user behaviors.

- **Graph Structure:** It consists of 8,405 user nodes.

- **Anomaly Definition:** Anomalous behavior is defined as "screen flooding"—posting two consecutive posts within a specific time window (e.g., 60 seconds). Users who engage in such behavior more than 5 times are labeled as "Suspicious", while others are labeled as "Benign".

- **Features:** The raw feature vectors consist of location information from posts and Bag-of-Words features derived from the text.

## E.2. Tolokers

Collected from the crowdsourcing platform Toloka (Platonov et al., 2023), this dataset is used to detect the reliability of crowd workers.

- **Graph Structure:** Nodes represent crowd workers who participated in at least one of 13 selected projects. An edge is established between two workers if they have collaborated on the same task.

- **Task Objective:** To predict which workers have been banned from specific projects due to poor performance or rule violations.

- **Features:** Node features are constructed based on the workers' profile information and statistical data regarding their historical performance.

## E.3. Questions

Sourced from the Q&A website Yandex Q (Platonov et al., 2023), focusing on the "Medicine" topic section.

- **Graph Structure:** Nodes represent users. An edge is created if one user answered a question posed by another user within a one-year period (September 2021 to August 2022).

- **Task Objective:** Churn prediction, i.e., identifying which users remain active at the end of the period.

- **Features:** Node features are the mean FastText embeddings of the user's description text. Additionally, a binary feature indicates whether the user has not provided a description.

## E.4. Yelp

Used for Spam Review Detection on Yelp.com (Rayana & Akoglu, 2015), aiming to identify anomalous reviews intended to unfairly promote or maliciously demote businesses. The graph structure includes three types of heterogeneous edges:

- **R-U-R:** Connects reviews posted by the same user.

- **R-S-R:** Connects reviews that give the same star rating to the same product.

- **R-T-R:** Connects reviews posted for the same product in the same month.

## E.5. Amazon

Designed to identify users writing paid fake reviews in the "Musical Instruments" category on the Amazon platform (McAuley & Leskovec, 2013). The graph defines three types of user interaction relations:

- **U-P-U:** Connects users who have reviewed at least one common product.

- **U-S-U:** Connects users who have given at least one identical star rating within a week.

- **U-V-U:** Connects users with top 5% mutual review text similarity.

## E.6. Elliptic

A large-scale Bitcoin transaction network dataset (Weber et al., 2019) used for Anti-Money Laundering (AML) and illicit activity detection.

- **Graph Structure:** Contains over 200,000 Bitcoin transaction nodes and 234,000 directed payment flow edges.

- **Features:** Each node possesses 166-dimensional features, some of which are constructed based on time steps.

- **Labels:** Bitcoin transactions are mapped to real-world entities categorized as either licit (such as exchanges and miners) or illicit (including scams, malware, ransomware, and Ponzi schemes).

### E.7. DGraph

Provided by Finvolution Group (Huang et al., 2022), DGraph is a large-scale dynamic graph dataset derived from a financial institution offering personal loan services.

- **Graph Structure:** The dataset comprises approximately 3 million nodes and 4 million dynamic edges. Nodes represent user accounts, and edges primarily denote "Emergency Contact" relationships (where one account sets another as an emergency contact).

- **Anomaly Definition:** Anomalies correspond to users with illicit financial behaviors. Specifically, for accounts with borrowing records, nodes representing users with a history of *overdue repayment* are labeled as anomalous.

- **Features:** Each node is associated with 17 features extracted from user profile information.

## F. Evaluation Metrics

To provide a comprehensive assessment of detection performance across different dimensions, we employ three standard metrics: AUC, AP, and Rec. Given the severe class imbalance inherent in graph anomaly detection tasks, as anomalous samples are exceedingly scarce, these metrics quantify the model's ranking capability and identification precision regarding anomalous nodes from distinct perspectives.

**AUC (Area Under the Receiver Operating Characteristic Curve):** AUC is a threshold-independent metric used to gauge binary classification performance. Mathematically, it is equivalent to the probability that the model ranks a randomly selected anomalous node higher than a randomly selected normal node. By integrating the trade-off between the True Positive Rate (TPR) and the False Positive Rate (FPR) across all possible decision thresholds, AUC provides a robust evaluation of the model's global ranking capability. It remains unaffected by the selection of specific classification thresholds, making it particularly suitable for assessing the model's discriminative power over the entire distribution.

**AP (Average Precision):** AP serves as an approximation of the area under the Precision-Recall (PR) curve, computed as the weighted mean of precision achieved at each threshold. Compared to AUC, AP is significantly more rigorous and sensitive in scenarios characterized by extreme class imbalance. It places greater emphasis on the prediction quality regarding the minority class (anomalous nodes). Consequently, AP accurately reflects the model's capacity to identify a higher volume of anomalies while maintaining a low false positive rate.

**Rec (Recall at k):** Rec@k is a ranking-oriented retrieval metric designed to evaluate the model's effectiveness within the "Top-$k$" recommendation list. In our experiments, we set $k$ equal to the total number of ground-truth anomalies in the dataset. This metric calculates the proportion of total ground-truth anomalies that are successfully retrieved within the top-$k$ nodes predicted as most suspicious. Rec@k intuitively measures the model's ability to "push" anomalous samples to the top of the ranking list. It holds significant practical value for resource-constrained detection scenarios, where only a limited number of high-risk nodes can be manually verified.

## G. Detailed Description of Baselines

This section provides a detailed elaboration of the three categories of baseline methods used for comparative evaluation: General Graph Neural Networks, specialized Graph Outlier Detectors, and Data Augmentation methods.

### G.1. General Graph Neural Networks

These methods represent the foundational architectures for graph representation learning, used to verify the model's performance in general feature extraction capabilities.

**GCN (Graph Convolutional Networks) (Kipf & Welling, 2017):** This is the pioneering work in graph deep learning. GCN defines an efficient hierarchical propagation rule by simplifying Chebyshev polynomials. By aggregating first-order neighborhood information to update node representations, it effectively captures the local homophilic structure of the graph.

**SGC (Simple Graph Convolution) (Wu et al., 2019):** As a lightweight variant of GCN, SGC assumes that non-linear activation functions between GCN layers are not the primary influencing factor. By removing inter-layer non-linear transformations, it simplifies multi-layer graph convolution into a single-layer linear transformation. This design significantly

enhances computational efficiency on large-scale graphs while retaining the ability to capture high-order neighborhood information.

**GIN (Graph Isomorphism Network) (Xu et al., 2019):** GIN is a class of GNNs with high expressive power. By introducing an injective aggregation function, GIN can distinguish different graph structures, generating permutation-invariant embeddings for nodes.

**GraphSAGE (Graph Sample and Aggregate) (Hamilton et al., 2017):** GraphSAGE proposes an inductive learning framework that addresses the problem of embedding generation for unseen nodes. This method efficiently generates node representations by sampling a fixed number of neighbor nodes and fusing neighborhood features using specific aggregation functions.

**GAT (Graph Attention Networks) (Veličković et al., 2018):** GAT introduces the attention mechanism to graph-structured data. Unlike the isotropic aggregation of GCN, GAT adaptively assigns weights to different neighbors by calculating attention coefficients between neighbor nodes. This allows the model to identify and focus on neighborhood structures that are more critical to the current node's task.

**GT (Graph Transformer) (Shi et al., 2021):** This model adapts the Transformer architecture not only for sequence data but also for graph structures. GT compensates for the lack of graph topological information by introducing positional or structural encodings and utilizes a global self-attention mechanism to capture long-range dependencies between nodes.

**RGCN (Relational Graph Convolutional Networks) (Schlichtkrull et al., 2018):** RGCN is an extension of GCN for multi-relational graph data. By assigning specific linear transformation matrices to each relation type, it explicitly distinguishes the influence of different edge types on node representations, effectively improving the model's generalization ability on sparse relational data.

**HGT (Heterogeneous Graph Transformer) (Hu et al., 2020):** HGT is a Transformer architecture designed specifically for heterogeneous graphs. HGT employs a meta-relation-aware parameterization strategy, enabling it to automatically learn the importance differences between diverse node and edge types, thereby extracting effective semantic information from complex heterogeneous interactions.

### G.2. Graph Outlier Detectors

These methods are specifically designed to detect anomalous patterns (e.g., structural anomalies, attribute anomalies) on graphs.

**GAS (GCN-based Anti-Spam) (Li et al., 2019):** GAS is designed for spam review detection. It utilizes a GCN architecture enhanced with an attention mechanism to capture complex interaction patterns. It effectively identifies spam behaviors and structural anomalies from the local context.

**DCI (Deep Cluster Infomax) (Wang et al., 2021):** DCI adopts a self-supervised learning paradigm that decouples node representation learning from anomaly detection. It utilizes Mutual Information Maximization to learn node embeddings and discovers latent community structures through a clustering mechanism, ultimately determining anomalies based on the deviation of nodes from cluster centers.

**PCGNN (Pick and Choose Graph Neural Network) (Liu et al., 2021):** Targeting the class imbalance problem prevalent in fraud detection, PCGNN reconstructs subgraphs by designing a Label-balanced Sampler. This balances the neighborhood distribution during training, thereby improving the model's recognition rate for the minority class (fraudulent nodes).

**GATSep (Zhu et al., 2020):** Addressing the poor performance of traditional GNNs on heterophilic graphs, GATSep introduces a separation mechanism for self-loops and neighborhoods. Unlike approaches that mix all information, GATSep processes node features and neighborhood features independently. This effectively preserves high-frequency anomaly signals during aggregation, avoiding the issue where anomalous node features are drowned out due to oversmoothing.

**BernNet (Bernstein Graph Convolutional Network) (He et al., 2021):** BernNet introduces spectral filters based on Bernstein polynomials. Unlike traditional low-pass filters, BernNet can learn frequency response functions of arbitrary shapes, enabling it to effectively capture high-frequency band-pass information containing anomaly signals.

**AMNet (Adaptive Multi-frequency GNN) (Chai et al., 2022):** AMNet further leverages spectral theory by stacking multiple BernNet units to construct adaptive filters. It can simultaneously capture low-frequency (smooth background) and

high-frequency (abrupt anomalies) components in graph signals, adaptively fusing features from different frequency bands to enhance detection accuracy.

**BWGNN (Beta Wavelet Graph Neural Network) (Tang et al., 2022):** BWGNN reveals that graph anomalies typically manifest as a "right-shifted" high-frequency phenomenon in the spectrum. The model utilizes Beta wavelets to construct band-pass filters with localized characteristics, specifically responding to these high-frequency anomaly features to precisely localize anomalies in densely connected graphs.

**GHRN (Graph Heterophily Reduction Network) (Gao et al., 2023):** GHRN addresses the interference caused by heterophily from a spectral domain perspective. By learning the graph structure and pruning cross-class edges (edges connecting nodes of different classes), it effectively removes structural noise components and enhances homophily, thereby improving GNN performance on anomaly detection tasks.

**CARE-GNN (CAmouflage-REsistant GNN) (Dou et al., 2020a):** CARE-GNN is a fraud detector designed for multi-relational graphs to resist camouflage. The model adaptively filters out camouflaged neighbor nodes and performs weighted aggregation for different relations, significantly enhancing detection robustness.

**H2-FDetector (Fraud Detector with Homophilic and Heterophilic Interactions) (Shi et al., 2022):** H2-FDetector is a heterogeneous graph neural network framework capable of handling both homophilic and heterophilic interactions. It designs dedicated components to automatically identify the nature of relations, supporting the aggregation of similar features via homophilic connections while capturing differentiated patterns (or negative correlations) via heterophilic connections.

**SpaceGNN (Multi-Space Graph Neural Network) (Dong et al., 2025):** SpaceGNN is a multi-space graph neural network designed for node anomaly detection (NAD) tasks with extremely limited labels, which integrates Euclidean and non-Euclidean space information via the Learnable Space Projection (LSP), Distance Aware Propagation (DAP), and Multiple Space Ensemble (MulSE) modules.

### G.3. Data Augmentation Methods

**DAGAD (Data Augmentation for Graph Anomaly Detection) (Liu et al., 2022):** DAGAD is an augmentation framework specifically designed for graph anomaly detection. It consists of three core modules: a generative augmentation module to enrich training samples, an information fusion module to merge original and augmented features, and an adapter learning module to balance normal and anomalous classes. This method effectively mitigates training difficulties caused by the scarcity of anomalous samples.

**GODM (Graph Outlier Detection via Latent Diffusion Models) (Liu et al., 2025):** This method introduces Latent Diffusion Models into the graph anomaly detection task. It aims to leverage the powerful distribution fitting capabilities of generative models to synthesize outlier nodes, thereby effectively alleviating the problem of anomaly sample scarcity.

## H. Analysis of Further Experiments

### H.1. Experiments on Large-scale Financial Graphs

Our experimental results are shown in Table 7.

*Table 7.* Performance comparison on five datasets including large-scale graphs (AUC, AP, and Rec in %).

| DATASET METRIC | WEIBO | | | TOLOKERS | | | QUESTIONS | | | ELLIPTIC | | | DGRAPH | | |
|---|---|---|---|---|---|---|---|---|---|---|---|---|---|---|---|
| | AUC | AP | REC | AUC | AP | REC | AUC | AP | REC | AUC | AP | REC | AUC | AP | REC |
| GCN | 98.11 | 93.48 | 89.34 | 74.69 | 42.88 | 42.06 | 69.81 | 12.54 | 16.99 | 82.68 | 22.23 | 27.61 | 75.85 | 3.99 | 7.05 |
| GRAPHSAGE | 96.54 | 89.25 | 86.17 | 79.42 | 48.65 | 46.42 | 71.69 | 17.63 | 21.10 | 85.31 | 37.52 | 36.20 | 75.63 | 3.76 | 6.97 |
| BWGNN | 98.29 | 92.72 | 84.73 | 80.15 | 49.65 | 47.35 | 69.47 | 16.24 | 18.63 | 84.32 | 22.56 | 26.50 | 76.26 | 4.01 | 7.52 |
| GHRN | 97.21 | 92.67 | 88.18 | 79.80 | 49.50 | 48.29 | 68.24 | 16.24 | 18.63 | 85.36 | 24.01 | 30.29 | 76.14 | 3.99 | 7.53 |
| GODM | 99.57 | 97.54 | 93.08 | 83.46 | 52.95 | 52.96 | 76.84 | **20.48** | 24.66 | **89.77** | 43.92 | 53.92 | 79.05 | 4.49 | 7.80 |
| **GOCM** | **99.62** | **97.92** | **93.94** | **83.69** | **55.39** | **53.86** | **77.08** | 19.39 | 24.04 | 88.40 | **46.02** | **54.01** | **79.26** | **4.64** | **8.19** |

On the challenging financial datasets Elliptic (large-scale) and DGraph (ultra-large-scale), GOCM demonstrates critical performance advantages. Although the AUC on Elliptic is slightly lower than that of GODM, GOCM achieves superior results in AP and Rec, which are metrics that prioritize the prediction quality of positive samples (i.e., outliers).

On DGraph, which possesses the largest node scale, GOCM achieves improvements across all metrics. This is attributed to the OC model employed by GOCM. Compared to GODM, which requires dozens of iterative diffusion sampling steps, GOCM generates high-quality samples directly from noise via a single-step mapping. This mechanism not only avoids error accumulation caused by long-trajectory sampling on large-scale graphs but also offers better numerical stability. Consequently, it generates more discriminative samples within complex financial risk control graphs, effectively enhancing the model's ability to identify concealed anomalies.

## H.2. Quality and Distribution of Synthesized Anomalies

To evaluate the quality of the generated samples, we visualize the feature density distribution of both the synthesized anomalies and the real anomalies in the original feature space.

As illustrated in Fig. 5, the overall distribution of the anomalies synthesized by GOCM closely aligns with the distribution of the real anomalies.

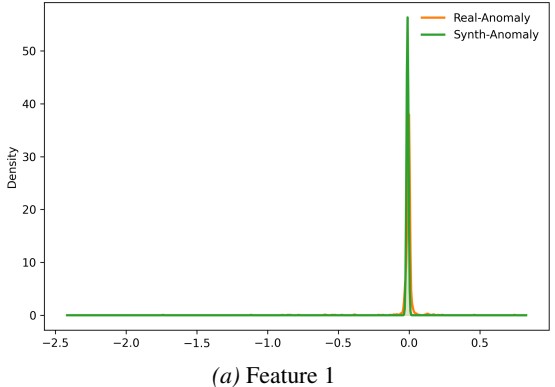
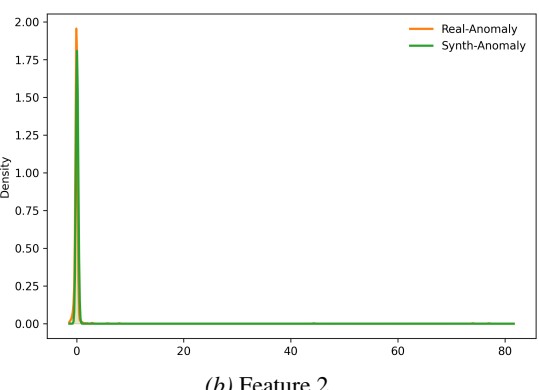

*(a)* Feature 1                    *(b)* Feature 2

*Figure 5.* **Feature Density Distribution.** Comparison between the real anomalies and the anomalies synthesized by GOCM across two different features on the Weibo Dataset.

## H.3. Robustness to the Latent Representation

To evaluate the sensitivity of the OC mechanism to the latent representations learned by MiVGAE, and to determine if the model remains effective with a simpler or limited-performance encoder, we tracked the downstream detection performance across varying MiVGAE training epochs.

Fig. 6 shows that even when relying on under-trained representations at early stages, our framework consistently outperforms the standard baseline (GraphSAGE: 79.42).

## H.4. Effectiveness of Edge Generation Mechanism

We validate the effectiveness of the edge structure generation mechanism used in GOCM. We compare our approach against two baselines: randomly assigning edges between generated nodes; generating edges based on the degree distribution heuristics of the original graph.

Table 8 presents the performance comparison on the Amazon dataset using RGCN as the backbone detector. The results demonstrate that our generation mechanism outperforms both random and degree-based heuristics across all three metrics.

*Table 8.* Effectiveness of Edge Generation Mechanism (AUC, AP, and Rec in %).

| METHOD | AUC | AP | REC |
|---|---|---|---|
| RANDOM | 92.13 | 68.85 | 66.30 |
| DEGREE-BASED | 93.40 | 71.11 | 69.02 |
| **GOCM** | **93.65** | **73.78** | **70.11** |

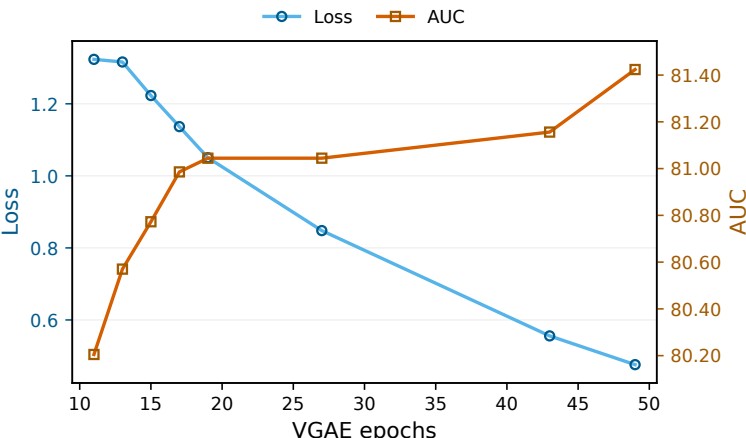

*Figure 6.* **Robustness Evaluation.** The evolution of downstream detection AUC (%) and the loss across varying MiVGAE training epochs on Tolokers Dataset.

## H.5. Performance under Extreme Label Scarcity

To test the robustness of our model against label scarcity, we varied the proportion of labeled anomaly nodes kept in the training set of the Weibo dataset (treating the remaining unselected anomalies as normal nodes).

As illustrated in Fig. 7, when only 1% of the anomaly labels are available, the performance of the standard GraphSAGE baseline collapses significantly to an AUC of 47. In contrast, our full framework remains highly robust, maintaining AUC of 95. This performance margin confirms that our generative mechanism provides effective and high-quality sample completion.

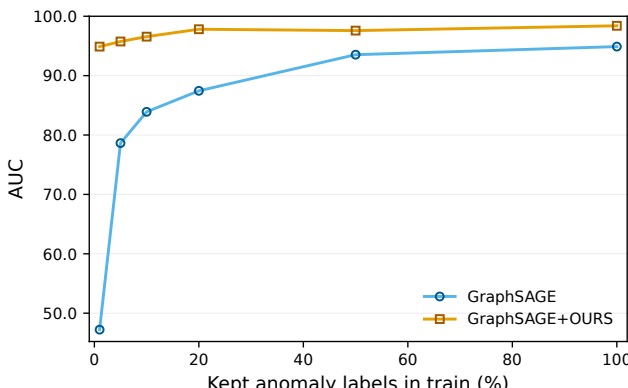

*Figure 7.* **Robustness to Label Scarcity.** Downstream detection AUC (%) on the Weibo dataset across varying proportions of kept anomaly labels in the training set.

## I. Analysis of Further Ablation Study

### I.1. Ablation Study Analysis

Our comprehensive ablation experimental results across multiple detectors and datasets are shown in Table 9.

In the ablation study, we observe that utilizing a standard One-step Consistency Model within the same VGAE latent space in GOCM, which underperforms relative to our OC model, still achieves performance nearly equivalent to that of GODM. This phenomenon is primarily attributed to two factors:

First, the generation process occurs in a low-dimensional and relatively smooth latent space. A single large-magnitude

*Table 9.* Performance comparison of different detectors under CM and OC methods (AUC, AP, and Rec in %).

| DETECTOR | METHOD | WEIBO | | | TOLOKERS | | | QUESTIONS | | | ELLIPTIC | | |
|---|---|---|---|---|---|---|---|---|---|---|---|---|---|
| | | AUC | AP | REC | AUC | AP | REC | AUC | AP | REC | AUC | AP | REC |
| GCN | CM | 99.52 | 97.43 | 93.25 | 78.56 | 48.32 | 47.66 | 74.53 | 15.30 | 23.01 | 83.26 | 26.13 | 32.77 |
| | OC | **99.62** | **97.92** | **93.94** | **78.81** | **49.33** | **47.75** | **74.68** | **15.52** | **23.28** | **83.92** | **26.80** | **34.43** |
| GAS | CM | 97.08 | 93.58 | 89.33 | 75.71 | 45.02 | 43.92 | 68.28 | 16.80 | 20.00 | **88.81** | 37.86 | 44.69 |
| | OC | **97.47** | **94.34** | **89.91** | **77.67** | **46.08** | **44.68** | **69.99** | **16.84** | **21.64** | 88.40 | **46.02** | **54.01** |
| GRAPHSAGE | CM | 98.94 | 91.48 | 88.76 | 81.59 | 50.93 | 49.22 | 75.95 | 17.64 | 23.28 | 85.76 | 43.70 | 42.38 |
| | OC | **98.97** | **94.25** | **89.72** | **81.68** | **52.42** | **50.10** | **77.08** | **19.39** | **24.04** | **86.09** | **48.22** | **50.13** |

denoising step is sufficient to push samples from the prior distribution close to the data manifold, rendering long integration trajectories unnecessary.

Second, the consistency training objective explicitly constrains denoising mappings at different noise levels to converge to the same original data point. This encourages the endpoint of multi-step diffusion trajectories to be well-approximated by a one-step mapping.

Since our evaluation metrics focus on downstream anomaly detection performance (AUC, AP, Rec) rather than absolute sample fidelity, capturing the decision boundary structure in the latent space is more critical than precisely matching the entire generative distribution. In this context, a fully trained one-step consistency model is sufficient to rival a 50-step diffusion model while possessing a significant advantage in efficiency.

## I.2. Ablation Study on Conditional Label

Since graph outlier detection suffers from extreme class imbalance, our generative model must be explicitly guided by the condition label to successfully synthesize the scarce anomaly nodes. To quantitatively validate the impact of this conditional mechanism, we designed a noise-injection ablation study. Specifically, we introduce continuous Gaussian noise of varying intensities (denoted by $\sigma_c$) into the condition signal during the synthesis phase, and then evaluate the downstream performance on the Tolokers dataset using GraphSAGE.

As shown in Table 10, compared to the clean condition ($\sigma_c = 0.00$), introducing even a small amount of noise leads to a drop in AUC, AP and Rec. This confirms that precise condition guidance is indispensable for the model.

*Table 10.* Performance on the Tolokers dataset using GraphSAGE across varying condition noise levels $\sigma_c$ (AUC, AP, and Rec in %).

| NOISE LEVEL ($\sigma_c$) | AUC | AP | REC |
|---|---|---|---|
| **0.00 (CLEAN)** | **82.01** | **52.95** | **51.35** |
| 0.02 | 81.87 | 52.17 | 51.09 |
| 0.10 | 81.62 | 52.02 | 49.95 |
| 0.20 | 81.82 | 51.94 | 50.47 |

## I.3. Isolating the Contribution of OC Mechanism

*Table 11.* Performance comparison between MiVGAE and MiVGAE + OC (AUC, AP, and Rec in %).

| MODEL | WEIBO | | | TOLOKERS | | | QUESTIONS | | | ELLIPTIC | | |
|---|---|---|---|---|---|---|---|---|---|---|---|---|
| | AUC | AP | REC | AUC | AP | REC | AUC | AP | REC | AUC | AP | REC |
| MIVGAE ONLY | 99.46 | 97.35 | 93.19 | 83.67 | 54.96 | 53.52 | 76.18 | 18.00 | 22.26 | 84.76 | 25.22 | 28.43 |
| **MIVGAE + OC** | **99.62** | **97.92** | **93.94** | **83.69** | **55.39** | **53.86** | **77.08** | **19.39** | **24.04** | **88.40** | **46.02** | **54.01** |

We compare our full framework (MiVGAE + OC) against a baseline that relies only on the latent representations learned by MiVGAE without any generative mechanism (MiVGAE only).

As shown in Table 11, the integration of the OC generative mechanism consistently leads to performance gains across all datasets and metrics.

