# OpenReview forum: "GOCM: Single-Step Graph Outlier Synthesis via Origin Consistency Model"
_ICML.cc/2026/Conference — ICML 2026 regular_

### Official Review · Reviewer_yRCY · 2026-03-07

**Soundness:** 2
**Presentation:** 2
**Significance:** 3
**Originality:** 2
**Overall Recommendation:** 4
**Confidence:** 3

**Summary:**

The paper proposes GOCM, a framework for supervised graph outlier detection that addresses class imbalance by synthesising additional outlier nodes. The main idea is to replace multi-step diffusion-based sampling with a single-step consistency-style generator using a new Origin Consistency (OC) mechanism that attempts to infer the data origin from a local time-interval segment of a probability-flow trajectory rather than enforcing pointwise consistency. The method operates in a latent space learned by a (heterogeneous) MiVGAE encoder, then decodes generated latent outliers back into node attributes and edges, including relation types for heterogeneous graphs. Empirically, the method shows improved detection performance over baselines and faster generation compared to iterative diffusion methods.

**Compliance With Llm Reviewing Policy:**

Affirmed.

**Final Justification:**

While the primary limitation: "quality of the generated instances and potential failure cases, for instance, under extreme scarce outlier scenarios." is still left unaddressed and i believe it is critical in understanding the efficacy of the method, the authors response has satisfactorily addressed my other concerns. I would recommend a weak accept.

**Key Questions For Authors:**

1. How does performance change under extreme label scarcity and varying imbalance (e.g., varying number of labelled anomalies during training)?
2. Do the authors have evidence that synthetic outliers are "realistic" beyond improving detector metrics (e.g., latent-space distributions, nearest-neighbor analysis to real outliers)?
3. How many synthetic nodes D_syn were generated and used for downstream training?
4. Could the authors please provide the computational cost involved with GOCM compared to baselines?

**Limitations:**

The paper lacks a detailed discussion on the quality of the generated instances and potential failure cases, for instance, under extreme scarce outlier scenarios.

**Strengths And Weaknesses:**

Strengths:
1. The focus on single-step generation is highly relevant for large graphs and efficiency-demanded scenarios, as prior diffusion-based augmentation can be slow.
2. The proposed GOCM is detector-agnostic in spirit, where it can improve many downstream detectors.
3. Extensive evaluations on diverse datasets demonstrate the efficacy of the method. While the improvements are not significant over competitive methods like GODM, it presents a greater speed up in generation.

Weaknesses:
1. The method assumes a linear trajectory x_t=(1-t) x_0 + tx_1 (lines 156-157). This can be a strong modelling choice, and it is not shown why this is the right inductive bias for graph latent distributions produced by MiVGAE, or whether the OC mapping remains valid under non-linear flows.
2. While GOCM demonstrates effective empirical performance on benchmarks, it lacks a comprehensive qualitative study of the quality and effectiveness of the synthetic outliers (e.g., embedding distributions). It is difficult to understand if the generated samples are truly outliers or fully out-of-distribution.
3. In lines 320-323, it is mentioned that the decoder predicts edge existence and relation type, but the evaluation focuses on node-level detection metrics. There is no strong evidence that the synthesized edges preserve graph-level statistics (degree distribution, homophily, temporal patterns etc.). If edges are unrealistic, the augmentation might inadvertently change message passing behaviour in ways unrelated to real outliers.
4. In the ablation study in Appendix H.2, it suggests that even a standard one-step consistency model in the same latent space can nearly match the multi-step diffusion baseline, implying that the biggest win might be latent-space smoothing rather than the OC mechanism itself. If so, the novelty claim for OC requires stronger evidence, for example, matched-capacity comparisons where CM and OC have the same compute and same latent encoder, or training stability.
5. Based on Figure 3, it shows that generation is faster than diffusion at inference, but the pipeline includes training MiVGAE and OC, plus decoding and integrating synthetic nodes/edges, and then re-training the downstream detector. It is not clearly how total wall-clock cost compares against baselines and the overall computational cost involved.
6. Since the method involves multiple hyperparameters, a detailed hyperparameter sensitivity analysis is missing (e.g., the weights w, beta, t,s).
7. The overall clarity can be improved by providing algorithms for the training and inference stage.

---

> ### Author Rebuttal · Authors · 2026-03-30
>
> Dear Reviewer yRCY
>
> We deeply appreciate your comprehensive and constructive review. Your detailed questions are highly insightful.(Please see the anonymous link for added figures/tables: https://anonymous.4open.science/r/rebuttal_4-CB96)
>
> **Q1: Regarding the linear trajectory assumption and its applicability to complex non-linear graph flows.**
> **A1:** According to Reflow theory (Liu et al., 2023), any complex non-linear target distribution can be exactly reconstructed by marginalizing linear conditional paths. The linear assumption only defines the "straight-line displacement" of individual sample pairs; the aggregated global probability flow still perfectly fits complex non-Euclidean manifolds. While the OC objective can theoretically adapt to non-linear flows by modifying the integral term, non-linear trajectories introduce high curvature, which would cause massive truncation errors during single-step generation. Thus, linear paths are the optimal mathematical choice for our one-step framework.
>
> **Q2 & Q9: Qualitative evidence concerning the realism and feature distribution of synthetic outliers.**
> **A2&9:** We extracted the features of both real anomalies and GOCM-synthesized anomalies and visualized their feature density distributions, which show that the synthesized samples closely align with the true anomaly density. (please refer to the anonymous link)
>
> **Q3: The impact of synthetic edges on graph-level statistics and message-passing behavior.**
> **A3:** Although edge generation utilizes a concise mechanism, our experiments reveal that it significantly outperforms random generation or degree-based heuristics. This suggests our straightforward projection effectively extracts discriminative connectivity signals from the MiVGAE latent space, ensuring the generated edges are realistic for downstream message-passing.
>
> **Q4: Distinguishing GOCM's novelty from the representation effects of latent-space smoothing.**
> **A4:** To isolate OC's contribution, we conducted the comparisons. First, comparing a "VGAE-only" baseline with our full framework confirms OC provides independent gains. Second, our standard "CM" baseline already operates within the exact same frozen latent space and compute capacity as OC. Experiments (see anonymous link) show OC consistently outperforms standard CM across three detectors, proving the gains stem directly from the OC mechanism.
>
> **Q5 & Q11: Analysis of the end-to-end computational cost and practical wall-clock efficiency.**
> **A5&11:** We have measured the overall computational cost (FLOPs) and actual wall-clock time. GOCM is significantly more efficient than the multi-step diffusion baseline (GODM). We measured the total computational cost: GOCM requires 2041.30 GFLOPs (Weibo) and 11090.16 GFLOPs (Questions), compared to GODM's 2819.55 and 14825.61 GFLOPs, respectively. Similarly, our total wall-clock time on Questions is only 71.13s (vs. 172.35s for GODM). The efficiency gained from single-step generation completely eclipses the minor generative training overhead.
>
> **Q6: Sensitivity analysis of hyperparameters.**
> **A6:** We clarify that $t$ and $s$ are not tuned hyperparameters; they are randomly sampled time-steps during each forward pass. The weight $w$ is a mathematically derived adaptive factor, not a heuristic. For $\beta$, we followed standard VAE regularization settings for fair comparison. Moreover, our new sensitivity analysis shows that the model's performance remains highly stable across various warm-up factors ($\lambda$), proving the framework's inherent robustness without requiring extreme hyperparameter fine-tuning.
>
> **Q7: Algorithms for training and inference.**
> **A7:** To improve clarity, we will include detailed algorithm pseudocode for the Origin Consistency Model training and inference stages in the revised appendix. This is also available via the anonymous link.
>
> **Q8: Model performance under conditions of extreme label scarcity.**
> **A8:** To test label-scarcity robustness, we varied the labeled anomaly ratio on Weibo (others treated as normal). At 1% labels, GOCM remains robust (0.95 AUROC) while GraphSAGE collapses (0.47). This confirms that our OC-based generative mechanism provides sample completion, effectively offsetting the performance degradation caused by extreme label scarcity (see anonymous link).
>
> **Q10: Number of synthetic nodes $D_{\text{syn}}$ generated**
> **A10:** By default, the number of generated synthetic nodes is set to match the number of labeled anomaly nodes in the training set.
>
> We will carefully incorporate all the aforementioned discussions, newly added experimental results, and visualizations into the revised version of our manuscript.

---

> > ### Author Rebuttal · Reviewer_yRCY · 2026-04-02
> >
> > Thanks to the authors for the detailed reply. I have no more questions and will update the score accordingly.

---

### Official Review · Reviewer_ZB3b · 2026-03-11

**Soundness:** 2
**Presentation:** 2
**Significance:** 3
**Originality:** 3
**Overall Recommendation:** 4
**Confidence:** 4

**Summary:**

This paper proposes GOCM (Graph Outlier Synthesis via Origin Consistency Model), a single-step generative framework for graph outlier detection. The key contributions are: (1) an Origin Consistency (OC) mechanism that enables one-step denoising from noise to data origin, and (2) MiVGAE, a multi-relation variational graph autoencoder for heterogeneous graphs. The method is evaluated on 7 benchmark datasets including large-scale financial graphs.

**Compliance With Llm Reviewing Policy:**

Affirmed.

**Final Justification:**

Thank you for the detailed clarification. As this is a well-motivated and well-designed model. I am inclined to maintain my relatively positive score.

**Key Questions For Authors:**

## Questions for Authors

1.We are concerned about the scalability and computational cost of this the proposed method, especially used in large scale graph data, but the number of nodes of Tolokers and Questions are not directly found in this paper.
2. What is the actual computational cost of computing the Jacobian terms in Eq. (7) during training ? Does this offset the claimed efficiency advantage?

**Limitations:**

The relation between severness of class imbalance and different GNN methods is not adequately discussed.

**Strengths And Weaknesses:**

## Strengths

- **Efficiency**: The proposed single-step generation (1-NFE) achieves significant speedup compared to iterative diffusion models like GODM, as shown in Figure 3.
- **Heterogeneous graph handling**: MiVGAE effectively processes multi-relational graphs through intra-relation convolution and cross-relation fusion.
- **Strong empirical results**: GOCM achieves competitive or state-of-the-art performance on multiple benchmarks, including large-scale datasets like DGraph.

### Weaknesses

**1. Reproducibility Concerns**
- The "warm-up factor" $\lambda(k)$ in Appendix B.6 is crucial for training stability but lacks sensitivity analysis.

**2. Incomplete Experimental Validation**

- **Missing critical ablation**: No fair comparison between "Standard CM + MiVGAE" vs "OC + MiVGAE" to isolate the contribution of the OC mechanism itself.
- **Conditional generation not validated**: While the model uses label $y$ as condition (Fig. 2), ablation studies on the impact of conditioning are absent.

---

> ### Author Rebuttal · Authors · 2026-03-30
>
> Dear Reviewer ZB3b
>
> Thank you for your rigorous evaluation. Your insights have significantly strengthened our empirical validation. (Please see the anonymous link for added figures/tables: https://anonymous.4open.science/r/rebuttal_3-BBE7)
>
> **Q1 The "warm-up factor" in Appendix B.6 is crucial for training stability but lacks sensitivity analysis.**
> **A1:** We conducted a sensitivity analysis on $\lambda$ using the Weibo dataset (see anonymous link). The results demonstrate that the mean performance metrics remain stable across various assigned values. This proves our model's performance is robust and not overly sensitive to the specific choice of the warm-up factor.
>
> **Q2 No fair comparison between "Standard CM + MiVGAE" vs "OC + MiVGAE" to isolate the contribution of the OC mechanism itself.**
> **A2:** We completely agree that isolating the generative mechanism under fair conditions is essential. Specifically, the baseline denoted as "CM" refers to a standard one-step Consistency Model operating within the exact same frozen VGAE latent space as our OC model. To further substantiate this, we have provided experiments (see anonymous link) in the anonymous link comparing "CM" and "OC" across three different downstream detectors (GCN, GAS, and GraphSAGE). These added results consistently demonstrate that OC provides independent and robust performance gains over the standard CM, regardless of the detector.
>
> **Q3 While the model uses label as condition (Fig. 2), ablation studies on the impact of conditioning are absent.**
> **A3:** Since graph outlier detection suffers from extreme class imbalance, our generative model must be explicitly guided by the condition label $\mathbf{y}$ to successfully synthesize the scarce anomaly nodes. To quantitatively validate this conditional mechanism, we designed a noise-injection ablation study (see anonymous link). By introducing continuous noise of varying intensities ($y_{\text{noisy}} = y + \sigma$) into the condition signal, we evaluated the downstream performance on the Tolokers dataset using GraphSAGE. The results clearly show a monotonic degradation in detection metrics as the condition noise level increases. This confirms that precise condition guidance is indispensable for the model to synthesize high-fidelity outliers.
>
> **Q4 We are concerned about the scalability and computational cost of this the proposed method, especially used in large scale graph data, but the number of nodes of Tolokers and Questions are not directly found in this paper.**
> **A4:** We apologize for omitting the exact node counts; Tolokers and Questions contain 519,000 and 153,540 nodes, respectively (Appendix E will be updated). Regarding computational cost, GOCM requires 2041.30 GFLOPs (Weibo) and 11090.16 GFLOPs (Questions). This is lower than the multi-step diffusion baseline, GODM (2819.55 and 14825.61 GFLOPs, respectively), proving our decoupled architecture controls overhead and scales excellently to large graphs.
>
> **Q5 What is the actual computational cost of computing the Jacobian terms in Eq. (7) during training ? Does this offset the claimed efficiency advantage?**
> **A5:** The actual computational overhead of the Jacobian terms in Eq. (7) is low. These expressions are used exclusively to construct the regression target with a stop-gradient applied, effectively avoiding time-consuming double backpropagation. Furthermore, employing the Jacobian-Vector Product (JVP) limits the additional overhead to a constant range. In our experiments on an RTX 3090, the total wall-clock time for the GOCM augmentation module on the Questions dataset is only 71.13s(compared to 172.35s for GODM), confirming its high practical efficiency.
>
> We will carefully incorporate all the aforementioned discussions, newly added experimental results, and visualizations into the revised version of our manuscript.

---

> > ### Author Rebuttal · Reviewer_ZB3b · 2026-04-04
> >
> > Thank you for the clarification. Some of my concerns have been addressed. However one of the key contribution in efficiency is not outstanding compared to other methods in large dataset. As this is a well-motivated and well-designed model. I am inclined to maintain my relatively positive score

---

> > > ### Author Response · Authors · 2026-04-07
> > >
> > > Dear Reviewer ZB3b,
> > >
> > > Thank you for your constructive follow-up and for maintaining your positive evaluation of our model's motivation and design.
> > >
> > > We acknowledge your valid point regarding the efficiency on large datasets. It is true that our generative framework introduces a higher upfront computational cost compared to lightweight, non-generative baselines. However, based on the task difficulty of modeling complex graph topologies under significant class imbalance, it is a trade-off made between generating high-fidelity structural anomalies and execution efficiency.
> > >
> > > In applications, decoupling the pre-training or augmentation phase from downstream tasks is a common method to balance model performance and computational overhead (Zhou et al., 2023). Specifically, GOCM functions as an independent data augmentation plug-in. Once the generative model undergoes its training and produces the synthetic outliers, this augmented data can be reused across downstream GNN detectors. In contrast, joint training methods require re-executing the entire pipeline when the downstream detection architecture is updated.
> > >
> > > Furthermore, to demonstrate our efficiency advantage within the family of generative methods, we compare the wall-clock time with diffusion-based models (GODM and DiffGAD), as shown in the table below:
> > >
> > > | Method | Weibo (s) | Questions (s) | DGraph (s) |
> > > | :--- | :--- | :--- | :--- |
> > > | DiffGAD (Li et al., 2025) | 135.24 | OOM | OOM |
> > > | GODM | 33.01 | 172.35 | 683.95 |
> > > | **GOCM (Ours)** | **23.48** | **71.13** | **413.68** |
> > > *(Note: OOM denotes Out of Memory)*
> > >
> > > We will explicitly incorporate this discussion regarding the efficiency trade-off, the added efficiency comparison table, along with the relevant references, into the revised manuscript.
> > >
> > > Thank you again for your valuable feedback.
> > >
> > > **References:**
> > > Zhou, M., Lu, J., Song, Y., and Zhang, G. Multi-stream concept drift self-adaptation using graph neural network. *IEEE Transactions on Knowledge and Data Engineering*, 35(12):12828–12841, 2023.
> > >
> > > Li, J., Gao, Y., Lu, J., Fang, J., Wen, C., Lin, H., and Wang, X. DiffGAD: A diffusion-based unsupervised graph anomaly detector. In *Proceedings of the Thirteenth International Conference on Learning Representations (ICLR 2025)*, 2025.

---

### Official Review · Reviewer_QSu8 · 2026-03-12

**Soundness:** 3
**Presentation:** 3
**Significance:** 2
**Originality:** 2
**Overall Recommendation:** 4
**Confidence:** 2

**Summary:**

This paper proposes Graph Outlier Synthesis via Origin Consistency Model (GOCM), a data augmentation framework for supervised graph outlier detection. The key idea is to replace diffusion-based anomaly synthesis with a single-step generative process based on a proposed Origin Consistency (OC) mechanism. Experimental results on several benchmark datasets show improvements in anomaly detection performance and significantly faster generation compared to diffusion-based baselines.

**Compliance With Llm Reviewing Policy:**

Affirmed.

**Final Justification:**

The authors have added all the analyses and experiments I previously requested.

**Key Questions For Authors:**

1. Does the OC formulation theoretically guarantee that the generated samples follow the true anomaly distribution?
2. Can the authors provide visualization or statistical analysis of the generated anomaly samples?
3. How sensitive is the method to the latent representation learned by MiVGAE? Would the OC model still work well with simpler encoders?
4. What is the anomaly ratio of the experimental datasets? Will the performance be affected by the anomaly ratio?

**Strengths And Weaknesses:**

**Strengths:**

1. The paper is well-written and organised.
2. The motivation is well-grounded.
3. The generation framework is efficient.

**Weaknesses:**

1. The framework may rely heavily on the MiVGAE latent space encoding. It is possible that the performance improvement mainly comes from this representation learning component rather than the OC generative mechanism itself. More analysis such as isolating MiVGAE or OC to see the contribution of each component is needed.
2. Although the paper claims that the synthesised anomalies improve downstream detection performance, it does not analyse the quality or diversity of generated samples. For example, it would be helpful to visualize the latent distribution of generated anomalies or measure their similarity to real anomalies.

---

> ### Author Rebuttal · Authors · 2026-03-30
>
> Dear Reviewer QSu8:
>
> We sincerely thank you for your thorough evaluation and highly constructive feedback. Your insightful questions have been immensely helpful in strengthening the completeness of our work.
>
> (Please see the anonymous link for added figures/tables: https://anonymous.4open.science/r/rebuttal_2-68B9)
>
> **Q1: The framework may rely heavily on the MiVGAE latent space encoding. It is possible that the performance improvement mainly comes from this representation learning component rather than the OC generative mechanism itself. More analysis such as isolating MiVGAE or OC to see the contribution of each component is needed.**
> **A1:** MiVGAE provides a structured latent space for the generative task, but its own learning capacity is limited, prompting us to add the OC mechanism for effective anomaly synthesis. To clearly isolate the contributions of each component, we conducted ablation experiments (see anonymous link). Comparing the baseline that solely relies on representation learning ("VGAE only") with our full framework ("VGAE + OC") shows that incorporating the OC generative mechanism leads to further performance improvements across all datasets.
>
> **Q2: Although the paper claims that the synthesised anomalies improve downstream detection performance, it does not analyse the quality or diversity of generated samples. For example, it would be helpful to visualize the latent distribution of generated anomalies or measure their similarity to real anomalies.**
> **A2:** We have plotted the feature density distribution graphs of both the synthesized anomalies and the real anomalies in the original feature space (please see the anonymous link). The visualization results demonstrate that the sample distribution synthesized by GOCM closely aligns with that of the real anomalies.
>
> **Q3: Does the OC formulation theoretically guarantee that the generated samples follow the true anomaly distribution?**
> **A3:** Yes. According to Reflow theory (Liu et al.,2023), the marginal vector field constructed by linear trajectories can exactly reconstruct the true distribution. While our parameterized denoiser acts as a computationally tractable approximation in practice, at the theoretical global optimum, our OC training objective is strictly equivalent to the ideal denoiser, which defines the true posterior distribution. Therefore, under the theoretical assumption of perfect convergence, sampling from the OC model mathematically guarantees sampling from the true anomaly data distribution.
>
> **Q4: Can the authors provide visualization or statistical analysis of the generated anomaly samples?**
> **A4:** As detailed in A2, we have plotted the feature density distributions comparing the synthesized anomalies with real anomalies in the feature space. The visual analysis confirms that the distribution of GOCM-generated samples is very close to the real anomaly distribution (available in the anonymous link).
>
> **Q5: How sensitive is the method to the latent representation learned by MiVGAE? Would the OC model still work well with simpler encoders?**
> **A5:** To evaluate OC's compatibility with limited-performance encoders, we tracked downstream AUC across varying MiVGAE training epochs. The provided curve (see anonymous link) shows that even when relying on severely under-trained representations at early stages, our framework consistently outperforms the standard baseline (GraphSAGE: 79.42). This confirms OC's robustness.
>
> **Q6: What is the anomaly ratio of the experimental datasets? Will the performance be affected by the anomaly ratio?**
> **A6:** The anomaly ratios of our experimental datasets are as follows: Weibo (10.3%), Elliptic (9.8%), Tolokers (21.8%), Amazon (9.5%), YelpChi (14.5%), Questions (3.0%), and DGraph (1.3%). The absolute performance of the models is indeed affected by the anomaly ratio. On datasets with relatively higher anomaly ratios (e.g., Weibo, Elliptic, and Tolokers), the baseline models exhibit better evaluation metrics. Conversely, on severely imbalanced datasets like Questions and DGraph, the baseline metrics are relatively lower.
>
> We will carefully incorporate all the aforementioned discussions, newly added experimental results, and visualizations into the revised version of our manuscript.

---

> > ### Author Rebuttal · Reviewer_QSu8 · 2026-04-03
> >
> > The authors fully addressed my concerns.

---

### Official Review · Reviewer_2q1Q · 2026-03-17

**Soundness:** 3
**Presentation:** 3
**Significance:** 3
**Originality:** 3
**Overall Recommendation:** 4
**Confidence:** 3

**Summary:**

GOCM is a single-step graph outlier synthesis framework designed to address class imbalance in graph anomaly detection. It utilizes an Origin Consistency model to map noise directly to the data origin for efficient generation, and employs a Multi-input Variational Graph Auto-Encoder to handle multi-relational graph structures within a latent space.

**Compliance With Llm Reviewing Policy:**

Affirmed.

**Final Justification:**

My concerns have been adequately addressed

**Key Questions For Authors:**

See above.

1. The standard one-step consistency model achieves performance that is nearly equivalent to the proposed OC model. Does this marginal performance gain justify the added mathematical and architectural complexity of the interval-based inference

**Limitations:**

Yes

**Strengths And Weaknesses:**

Strengths

1. The paper addresses a practical efficiency bottleneck by reducing the iterative sampling process to a single step

2. MiVGAE offers a reasonable structural approach to decouple and encode multi-relational heterogeneous graphs into a unified latent space

3. The empirical evaluation spans multiple datasets, including large-scale financial graphs like DGraph, showing computational cost reductions compared to iterative models

Weaknesses

1. The mathematical derivation of the interval-based origin inference relies strictly on the assumption of a linear probability flow trajectory. This assumption will oversimplify the complex and non-Euclidean manifold structures of real-world graph latent spaces

2. The regression target formulation introduces a stop-gradient operation. The paper lacks sufficient analysis of the potential gradient estimation bias introduced by this approximation

3. The method relies on manually designed stabilization factors to control gradient variance during early training. This reliance on carefully tuned heuristics contradicts the earlier claim that the method avoids extreme sensitivity to hyperparameters

4. The two-stage training strategy freezes the MiVGAE parameters during the consistency model training stage. This decoupled approach prevents the generative target from providing feedback to the structural encoder, which will lead to suboptimal latent representations for outlier synthesis

5. The edge structure generation relies on a simple concatenation and linear projection of conditioned latent representations. This simple mechanism struggle to synthesize complex high-order structural graph anomalies

---

> ### Author Rebuttal · Authors · 2026-03-30
>
> Dear Reviewer 2q1Q
>
> Thank you for your rigorous and constructive review. Your insightful questions have helped us further clarify the theoretical and empirical soundness of our work.
> (Please see the anonymous link for added figures/tables: https://anonymous.4open.science/r/rebuttal_1-0164)
>
> **Q1: The mathematical derivation of the interval-based origin inference relies strictly on the assumption of a linear probability flow trajectory. This assumption will oversimplify the complex and non-Euclidean manifold structures of real-world graph latent spaces.**
> **A1:** Your concern is highly insightful. However, according to Reflow theory (Liu et al., 2023), the marginal vector field constructed by linear trajectories can exactly reconstruct the true complex distribution. Furthermore, in our decoupled architecture, the complex non-Euclidean topological properties of the graph are first embedded into a continuous latent space by the MiVGAE. Our OC mechanism then operates strictly within this continuous space. While our parameterized denoiser acts as a computationally tractable approximation, at the theoretical global optimum, our OC objective is strictly equivalent to the ideal denoiser. Therefore, sampling from the OC model mathematically guarantees sampling from the true anomaly distribution without oversimplifying the original non-Euclidean graph structures.
>
> **Q2: The regression target formulation introduces a stop-gradient operation. The paper lacks sufficient analysis of the potential gradient estimation bias introduced by this approximation.**
> **A2:** Based on Theorem 6 (Appendix B) of the original Consistency Models paper, in the continuous-time limit, the gradient of the consistency training objective with the stop-gradient (`sg`) operator rigorously converges to the gradient of a well-defined continuous-time objective. Thus, the `sg` operation reliably guides convergence and efficiently avoids double backpropagation without introducing harmful gradient estimation bias.
>
> **Q3: The method relies on manually designed stabilization factors to control gradient variance during early training. This reliance on carefully tuned heuristics contradicts the earlier claim that the method avoids extreme sensitivity to hyperparameters.**
> **A3:** The stabilization factor $w(t, s)$ is mathematically derived. The constant $\varepsilon$ merely prevents division by zero, perfectly matching standard CM baselines, rather than being a dataset-specific heuristic. Moreover, our newly added sensitivity analysis (see anonymous link) demonstrates that downstream performance remains highly stable across a certain range of warm-up factor values ($\lambda$). This proves GOCM's inherent architectural robustness rather than a reliance on hyperparameter fine-tuning.
>
> **Q4: The two-stage training strategy freezes the MiVGAE parameters during the consistency model training stage. This decoupled approach prevents the generative target from providing feedback to the structural encoder, which will lead to suboptimal latent representations for outlier synthesis.**
> **A4:** We purposely froze the MiVGAE parameters during generative training to prevent representation collapse (following the LDM paradigm). While our two-stage training does not attain the theoretical global optimum, but graph topological priors are fragile, allowing generative gradients to backpropagate into the structural encoder risks distorting the original graph's priors, which would ultimately degrade downstream detection performance.
>
> **Q5: The edge structure generation relies on a simple concatenation and linear projection of conditioned latent representations. This simple mechanism struggle to synthesize complex high-order structural graph anomalies.**
> **A5:** Although the decoder utilizes a concise mechanism, our experiments (see anonymous link) reveal that it significantly outperforms random generation or degree-based heuristics. This suggests that a straightforward linear projection can effectively extract discriminative connectivity signals, provided the latent space is sufficiently informative and well-structured by MiVGAE.
>
> **Q6: The standard one-step consistency model achieves performance that is nearly equivalent to the proposed OC model. Does this marginal performance gain justify the added mathematical and architectural complexity of the interval-based inference.**
> **A6:** The classic Consistency Model framework is indeed highly effective. Our proposed OC model attempts to explore a theoretical bridge between Reflow and CM to better adapt the framework for our specific task. Empirically, our experiments across diverse datasets and downstream detectors (see anonymous link) confirm that the OC mechanism consistently and robustly outperforms standard CM.
>
> We will carefully incorporate all the aforementioned discussions, newly added experimental results, and visualizations into the revised version of our manuscript.

---

> > ### Author Rebuttal · Reviewer_2q1Q · 2026-04-03
> >
> > NA

---

### Decision · Program_Chairs · 2026-04-30

**Decision:**

Accept (regular)

**Comment:**

GOCM proposes a single-step generative framework for synthesizing outlier nodes to address class imbalance in supervised graph anomaly detection. The two contributions are (1) an Origin Consistency mechanism that derives a direct noise-to-origin mapping via interval-based inference on linear probability flows, and (2) MiVGAE, a multi-input VGAE for heterogeneous graphs. Evaluation spans seven GADBench datasets including large-scale financial graphs. All four reviewers recommend weak accept (4/4/4/4), and all acknowledged the rebuttal as fully resolving their concerns.

Strengths: substantial efficiency gain over multi-step diffusion baselines, competitive or SOTA results across benchmarks, and a reasonable architectural solution for multi-relational graphs.
Weaknesses: the linearity assumption is defended via ReFlow theory but its fit to the MiVGAE latent space is not fully demonstrated; the two-stage frozen-encoder design is pragmatic but theoretically suboptimal; and the OC-vs-standard-CM gap is modest on individual datasets (e.g., 99.52 vs. 99.62 AUC on Weibo), though more consistent across detectors in the added rebuttal experiments. The rebuttal added sensitivity analysis, distribution visualizations, label-scarcity robustness, a fair CM-vs-OC comparison, and computational cost measurements.

The contribution is clearly articulated, and the empirical gains, especially in the generative-augmentation regime, are consistent and practically useful.